# Depth over Fidelity in Fixed-Budget Noisy Evolution Strategies

**Sichen Wang**[1]   **Zhipeng Lu**[1 2]

## Abstract

Noisy evolution strategies under fixed evaluation budgets face a depth–fidelity trade-off: spending evaluations to denoise intra-generation rankings reduces the number of distribution updates the optimizer can execute. We argue for depth over fidelity and propose probabilistic elite membership (PEM), which replaces hard rank-based weights in evolution strategies with conditional expected rank weights that integrate over ranking uncertainty. PEM preserves the conditional mean update while reducing conditional update dispersion—a Rao–Blackwellization of the noisy rank-based step. We instantiate PEM via residual bootstrapping (RB-PEM) with capped per-generation overhead, complemented by an adaptive probe-and-switch mechanism for low-noise regimes. Across the COCO bbob-noisy suite and external tasks including RL policy search and hyperparameter optimization, RB-PEM achieves consistent gains in high-misranking, budget-constrained settings.

## 1. Introduction

Many contemporary machine learning workflows involve stochastic black-box optimization, where the objective is accessible only through noisy function evaluations. This regime arises in policy search with stochastic rollouts, simulation-based design and control, hardware-in-the-loop tuning, and hyperparameter optimization where training and validation induce intrinsic randomness. In such problems, the dominant constraint is often a strict cap on the number of oracle queries, i.e., a fixed-budget protocol. Consequently, evaluations spent on repeated measurement to denoise directly reduce the number of distinct candidates explored and the number of adaptive updates performed.

[1]Department of Engineering, Shenzhen MSU-BIT University, Shenzhen, China [2]Guangdong Laboratory of Machine Perception and Intelligent Computing, Shenzhen, China. Correspondence to: Zhipeng Lu <zhipeng.lu@smbu.edu.cn>.

*Proceedings of the $43^{rd}$ International Conference on Machine Learning*, Seoul, South Korea. PMLR 306, 2026. Copyright 2026 by the author(s).

Among black-box optimizers, rank-based evolution strategies (ES), in particular CMA-ES (Hansen & Ostermeier, 2001), are appealing because they require only relative comparisons of candidates and inherit invariance that make them robust across problem scalings. However, their reliance on intra-generation ranking makes them especially sensitive to evaluation noise: if the objective values of a sampled population are perturbed, the induced permutation of ranks changes, resulting misled updates using misranked "elites." In other words, noisy evaluations turn into selection noise that directly perturbs the update direction.

A large body of work on noisy evolutionary optimization advocates allocating additional intra-generation evaluations, such as uniform $k$-fold resampling and aggregation, sequential sampling / racing, and uncertainty-handling variants of CMA-ES, to stabilize ranks before updating (Jin & Branke, 2005; Rakshit et al., 2017; Beyer & Sendhoff, 2007; Hansen et al., 2009; Groves & Branke, 2018; Birattari et al., 2002). These approaches typically prioritize per-generation fidelity: approximate the noiseless ordering as closely as possible before applying the standard rank-based update.

However, under a fixed-budget protocol, fidelity comes with cost: evaluating each candidate multiple times reduces the number of generations inversely. This matters because the strength of CMA-ES comes from iterated distribution learning (mean, covariance, step-size adaptation) accumulated through many generations. In particular, update is mostly sensitive to the elite threshold due to truncation selection, where the borderline candidates have nearly indistinguishable objective values and achieving reliable selection via repeated evaluations can be more costly.

This paper argues for a complementary principle for fixed-budget noisy ES: when ranking uncertainty is high, it can be more sample-efficient to keep the per-generation evaluation cost low, thereby preserving optimization depth, and to incorporate uncertainty directly into the selection weights. Concretely, we introduce *probabilistic elite membership* (PEM): rather than applying hard rank weights from a single noisy ranking, we target the conditional expected rank weights given the sampled candidates. PEM can be viewed as a Rao–Blackwellization of the standard noisy rank-based update, which preserves the conditional mean update while reducing conditional update dispersion.

Computing PEM exactly would require extensive reevaluations per candidate, risking depth collapse. We therefore propose *residual-bootstrapped* PEM (RB-PEM), a practical estimator that uses only a *capped, additive* reevaluation overhead per generation. Each generation evaluates every candidate once and uses a small targeted reevaluation set to calibrate the local noise model, then extracts standardized noise residuals and inserts them into a pooled residual bank that is reused across generations. With this amortized residual pool, we can simulate many bootstrap rankings at essentially zero evaluation cost (Efron & Tibshirani, 1993) and estimate expected weights accurately while keeping per-generation cost close to one evaluation per candidate. Since pooling residuals can fail under distributional mismatch, we decompose the pool-to-target mismatch into four falsifiable online-measurable components and use them as runtime diagnostics.

Depth-over-fidelity is not universally optimal: when rankings are already reliable, smoothing can weaken selection pressure and any additional reevaluation becomes overhead. We therefore introduce a low-cost *probe-and-switch* mechanism using a small probing budget to dynamically choose between standard CMA-ES and RB-PEM.

Our main contributions are:

- We identify and formalize a *depth–fidelity* tradeoff for noisy rank-based evolution strategies under strict evaluation budgets, clarifying when evaluation-stage uncertainty reduction can be sample-inefficient.

- We introduce Probabilistic elite membership (PEM) as a principled selection-stage target, which can be viewed as a Rao–Blackwell variance reduction.

- We design residual-bootstrapped PEM (RB-PEM) with capped per-generation overhead and an amortized residual pool, and provide theory and runtime checks that quantify and diagnose residual pool mismatch.

- We propose the adaptive rule probe-and-switch to prevent negative transfer by reverting to standard CMA-ES in low-misranking regimes.

- Across the COCO bbob-noisy suite and diverse external tasks (RL policy search and noisy hyperparameter optimization), RB-PEM consistently improves fixed-budget performance in high-misranking regimes, substantiating the thesis that integrating uncertainty at the selection stage can be more sample-efficient than reducing it at the evaluation stage.

Section 3 formalizes the fixed-budget protocol, noisy ranking, and the depth–fidelity tension. Section 4 presents PEM, residual bootstrapping, and probe-and-switch; Section 5 provides a theoretical explanation based on conditional update dispersion; and Section 6 evaluates the resulting algorithms under strict budgets.

## 2. Related Work

**Noise handling in ES.** Most noisy-ES methods modify how candidates are observed before applying a deterministic rank rule. Uniform resampling, racing, and sequential sampling spend extra evaluations to stabilize ranks (Beyer & Sendhoff, 2007; Branke & Schmidt, 2004; Groves & Branke, 2018; Birattari et al., 2002); UH-CMA-ES and RA-CMA-ES adapt reevaluations during the run (Hansen et al., 2009; Uchida et al., 2024); PSA-CMA-ES increases population size (Nishida & Akimoto, 2018); and LRA-CMA-ES preserves generation count by attenuating CMA-ES learning rates to maintain signal-to-noise ratio (Nomura et al., 2025), which in our experiments can reduce the effective mean learning rate to about $0.04$. Surrogate-assisted ES and DTS-CMA-ES use learned objective/posterior models, including GP surrogates and expected recombination weights (Bajer et al., 2019; Krause, 2022). Noisy BO similarly integrates uncertainty through global GP models and Monte-Carlo acquisitions such as LogEI/qNEI (Balandat et al., 2020; Ament et al., 2023; Frazier, 2018); RB-PEM instead avoids global surrogate and acts directly on the rank-based ES update.

**Ranking and selection, and our position.** R&S also allocates noisy samples to compare alternatives (Kim & Nelson, 2007; Hong et al., 2021; 2022), with fixed-confidence, fixed-budget, and procedure-selection variants (Branke et al., 2007; Frazier, 2014; Pearce & Branke, 2017). R&S targets correct selection within a fixed alternative set, while CMA-ES uses populations as adaptive distribution-learning steps. RB-PEM is complementary to observation-stage noise handling: it intervenes at selection by replacing hard ranks with probabilistic weights. To our knowledge, this stage has remained unaddressed in the noisy single-objective ES literature. For top-$\mu$ truncation, $w_i^\star = \Pr(r_i \leq \mu \mid x_{1:\lambda})/\mu$, i.e., normalized elite-set membership; the conditional expected rank $\mathbb{E}[r_i \mid x_{1:\lambda}]$ is a different scalar and would discard the rank-weight map. For logarithmic CMA-ES weights, PEM estimates $\mathbb{E}[w(r_i) \mid x_{1:\lambda}]$ directly, so internal reorderings among top-ranked candidates are included.

## 3. Preliminaries

### 3.1. Optimization under Noisy Evaluations

We study fixed-budget stochastic black-box optimization. The goal is to minimize an unknown objective

$$\min_{x \in \mathbb{R}^d} f(x), \tag{1}$$

where $f : \mathbb{R}^d \to \mathbb{R}$ may be nonconvex and is accessed only through noisy function evaluations. More explicitly, each *oracle* call at a point $x \in \mathbb{R}^d$ returns a random scalar

$$y(x) = f(x) + \varepsilon(x), \tag{2}$$

where $\mathbb{E}[\varepsilon(x)] = 0$ and $\mathrm{Var}(\varepsilon(x)) = \sigma^2(x) < \infty$.

We will mostly assume that repeated calls at the same point $x$ yield i.i.d. samples, and all oracle calls are conditionally independent given their query locations.

## 3.2. Fixed-Budget Protocol

Fix a positive integer $B$ as an evaluation budget. An algorithm produces a sequence of query locations $(x_1, \ldots, x_B)$ adaptively, where each $x_t$ is a measurable function of history $\{(x_s, y_s)\}_{s=1}^{t-1}$ and internal randomness. After querying $x_t$ it observes $y_t \sim y(x_t)$ and consumes one unit of budget. The algorithm must terminate once the $B$-th evaluation is consumed and a recommendation $\hat{x}_B$.

For population-based evolution strategies, it is convenient to group oracle calls by "generation". At generation $t$ the algorithm proposes $\lambda$ candidates $(x_{t,1}, \ldots, x_{t,\lambda})$ and may allocate $K_{t,i} \geq 1$ evaluations to candidate $i$, producing samples $y_{t,i}^{(1)}, \ldots, y_{t,i}^{(K_{t,i})}$ with $y_{t,i}^{(j)} \sim y(x_{t,i})$ i.i.d. The total number of oracle calls is

$$\sum_t \sum_{i=1}^{\lambda} K_{t,i} \leq B, \tag{3}$$

i.e., any resampling, re-evaluation, probing, or aggregation step is counted within the same budget.

The *simple regret* of a returned recommendation is

$$r_B = f(\hat{x}_B) - f^\star, \qquad f^\star = \inf_{x \in \mathbb{R}^d} f(x), \tag{4}$$

and we can measure fixed-budget performance by the expected simple regret $\mathbb{E}[r_B]$ under the joint randomness of oracle noise and sampling randomization of the algorithm.

## 3.3. Noisy Evaluation and Misranking

We follow the standard CMA-ES sampling and ranking formalism specified in Appendix A which summarizes the definitions and mean-update relations (27)–(34) used throughout. Under the noisy oracle (Section 3.1), an algorithm does not observe $f(x_{t,i})$ directly. Instead, each evaluation returns

$$y_{t,i} = f(x_{t,i}) + \varepsilon_{t,i}, \qquad \mathbb{E}[\varepsilon_{t,i} \mid x_{t,i}] = 0, \tag{5}$$

with i.i.d. noise across repeated calls at the same $x_{t,i}$. The ranking is therefore induced by the noisy values $y_{t,i}$: let $\hat{\pi}_t \in S_\lambda$ satisfy

$$y_{t,\hat{\pi}_t(1)} \leq y_{t,\hat{\pi}_t(2)} \leq \cdots \leq y_{t,\hat{\pi}_t(\lambda)}. \tag{6}$$

Accordingly, the rank-weighted step used by the algorithm becomes the random quantity

$$\hat{z}_{w,t} := \sum_{j=1}^{\lambda} w(j) \, z_{t,\hat{\pi}_t(j)}, \tag{7}$$

and the resulting mean update is

$$m_{t+1} = m_t + \eta_m \sigma_t A_t \hat{z}_{w,t}. \tag{8}$$

We call an iteration *misranked* if the noisy ordering differs from the true ordering, i.e., $\hat{\pi}_t \neq \pi_t^\star$ where $\pi_t^\star$ is defined by sorting $f(x_{t,i})$. Equivalently, a misranking occurs whenever there exists a pair $(i, j)$ such that

$$f(x_{t,i}) \leq f(x_{t,j}) \quad \text{but} \quad y_{t,i} > y_{t,j}. \tag{9}$$

Our analysis quantifies how such misrankings bias the rank-based update $\hat{z}_{w,t}$ relative to the noiseless step $z_{w,t}$ in (32).

## 3.4. Mainstream Noise Handling under Fixed Budgets: Fidelity over Depth

In response to misranking in rank-based evolution strategies, common approaches replace each single noisy value $y_{t,i}$ with an *aggregated estimate*

$$\bar{y}_{t,i} := \text{Agg}(y_{t,i}^{(1)}, \ldots, y_{t,i}^{(K_{t,i})}), \qquad K_{t,i} \geq 1, \tag{10}$$

and then rank candidates by $\bar{y}_{t,i}$ instead of $y_{t,i}$. Here Agg may be the sample mean/median/trimmed mean, and $K_{t,i}$ may be fixed (uniform resampling) or adaptively chosen (uncertainty handling/sequential sampling). Let the *per-generation evaluation cost* be

$$C_t := \sum_{i=1}^{\lambda} K_{t,i}. \tag{11}$$

Under the fixed budget $B$ (Section 3.2), the number of completed distribution updates (*depth*) is

$$T(B) := \max\Big\{T \in \mathbb{N} : \sum_{t=0}^{T-1} C_t \leq B\Big\}. \tag{12}$$

If every candidate is evaluated exactly $k$ times (and aggregated), then $K_{t,i} \equiv k$, hence $C_t = k\lambda$, and the budget constraint immediately implies

$$T \leq \Big\lfloor \frac{B}{k\lambda} \Big\rfloor \approx \frac{B}{k\lambda}. \tag{13}$$

This bound is just budget accounting: each completed generation consumes $k\lambda$ oracle calls, so at most $\lfloor B/(k\lambda) \rfloor$ generations can be executed.

Uniform resampling improves *fidelity* because aggregation reduces effective noise. For intuition, suppose $\varepsilon(x)$ is conditionally sub-Gaussian with proxy variance $\sigma^2(x)$, and Agg is the sample mean over $k$ i.i.d. samples, so $\bar{y}_{t,i} = f(x_{t,i}) + \bar{\varepsilon}_{t,i}$ with $\bar{\varepsilon}_{t,i}$ sub-Gaussian of scale $\sigma(x_{t,i})/\sqrt{k}$. For a pair $(i, j)$ with true gap $\Delta_{t,ij} := f(x_{t,j}) - f(x_{t,i}) > 0$, we have

$$\Pr\left(\bar{y}_{t,i} > \bar{y}_{t,j} \mid x_{t,i}, x_{t,j}\right) = \Pr\left(\bar{\varepsilon}_{t,i} - \bar{\varepsilon}_{t,j} > \Delta_{t,ij}\right)$$

$$\leq \exp\left(-\frac{k \, \Delta_{t,ij}^2}{2\big(\sigma^2(x_{t,i}) + \sigma^2(x_{t,j})\big)}\right). \tag{14}$$

Thus, increasing $k$ can reduce pairwise misranking exponentially in $k$, *but only through the squared gap* $\Delta_{t,ij}^2$. To push the above probability below a target $\delta$, it requires that

$$k \; = \; \Omega\left(\frac{\sigma^2(x_{t,i}) + \sigma^2(x_{t,j})}{\Delta_{t,ij}^2} \log\frac{1}{\delta}\right). \qquad (15)$$

The bottleneck is that the most consequential comparisons for rank-$\mu$ updates are those near the truncation boundary ($r \approx \mu$), where gaps are typically small. As the population concentrates, these gaps can shrink and make $k$ required for a fixed fidelity target potentially large by (15). The above intuition indicates that robust aggregators can mitigate heavy tails and outliers, but they still require multiple oracle calls per candidate to reduce ranking uncertainty.

### 3.5. Depth over Fidelity: A Fixed-Budget Principle

Sections 3.3 and 3.4 reckon that the mainstream noise handling literature largely takes a *fidelity-over-depth* stance: spend additional intra-generation evaluations (resampling) to make the induced ordering close to the noiseless one, then apply the usual rank-based update. The key question in fixed-budget optimization is thus: *how much progress per resampling is gained by increasing fidelity?*

Extra evaluations reduce misranking probability, but the returns can be sharply diminishing. Indeed, by equation (13) and (15), the comparisons that matter most for rank-$\mu$ updates are those near the elite threshold ($r \approx \mu$), where $\Delta_{t,ij}$ is typically small, since as the population concentrates these gaps shrink. Thus, the $k$ needed for a fixed fidelity target can grow, collapsing depth through $T \propto 1/k$.

This exposes a fixed-budget tension: fidelity-first schemes can spend substantial budget resolving near threshold to approximate a deterministic top-$\mu$ decision, but the resulting loss in depth removes opportunities for iterated adaptation (mean/covariance/step-size) that require many generations to accumulate. Under strict budgets, this can be a losing trade: resolving near threshold by driving misranking probability down may require a large multiplicative increase in evaluations per candidate, which collapses the optimization trajectory length.

This motivates a paradigm shift: instead of denoise-then-rank, we aim to integrate uncertainty at the selection stage while keeping generations cheap. Concretely, if we have a *residual bootstrapping* approach that uses one base evaluation per candidate plus a small (capped) number $K_t$ of targeted extra calls, yielding by (11)

$$C_t = \lambda + K_t, \qquad K_t \leq K_{\max} \ll \lambda, \qquad (16)$$

then the depth scales as

$$T \approx \frac{B}{\lambda + \mathbb{E}[K_t]} \approx \frac{B}{\lambda} \quad \text{when } \mathbb{E}[K_t] \ll \lambda. \qquad (17)$$

In practice, instead of denoising each $y_{t,i}$ until the ordering is reliable, we propagate uncertainty through selection by replacing hard elite membership with probabilistic weights $w_i^\star = \mathbb{E}[w(r_i) \mid x_{1:\lambda}]$, and we allocate extra evaluations only when they have high marginal value near the threshold, maintaining $C_t = \lambda + K_t$ with $\mathbb{E}[K_t] \ll \lambda$. Sections 4 and 5 develop an implementable mechanism and show how this depth-preserving treatment of uncertainty translates into better fixed-budget performance.

## 4. Method: PEM, Residual Bootstrapping, and Probe-and-Switch

Following the depth-over-fidelity spirit, our method implements *selection-stage uncertainty integration*: keep the baseline per-generation cost close to one evaluation per candidate, and spend a small capped number of additional evaluations to estimate how ranking noise should be integrated into the rank-weighted update. Concretely, we (i) define a principal target update via *probabilistic elite membership* (PEM), (ii) estimate PEM by *residual bootstrapping* with per-generation cost $C_t = \lambda + K_t$ and $K_t \leq K_{\max} \ll \lambda$, and (iii) avoid unnecessary overhead in low-misranking regimes following a *probe-and-switch* rule.

### 4.1. Probabilistic Elite Membership (PEM)

For a generation $t$ and the sampled candidates $x_{t,1:\lambda}$, noisy values $y_{t,1:\lambda}$ induce random ranks $r_{t,i} = \text{rank}_i(y_{t,1:\lambda})$ (29). Standard rank-based CMA-ES updates use deterministic weights $w(r_{t,i})$, therefore the update direction is random even conditional on $x_{t,1:\lambda}$ (Section 3.3).

We define the PEM of candidate $i$ as the conditional expectation of its rank weight given candidates:

$$w_{t,i}^\star \; := \; \mathbb{E}[w(r_{t,i}) \mid x_{t,1:\lambda}]. \qquad (18)$$

For the canonical top-$\mu$ truncation weights $w(r) = \frac{1}{\mu}\mathbf{1}\{r \leq \mu\}$, this has the probabilistic interpretation

$$w_{t,i}^\star \; = \; \frac{1}{\mu} \Pr(r_{t,i} \leq \mu \mid x_{t,1:\lambda}),$$

i.e., candidates near the truncation receive fractional membership proportional to their probability of being selected. More generally, with $w_k := w(k)$, $\delta_k := w_k - w_{k+1}$, and $p_{t,i,k} := \Pr(r_{t,i} \leq k \mid x_{t,1:\lambda})$,

$$w_{t,i}^\star = w_\lambda + \sum_{k=1}^{\lambda-1} \delta_k \, p_{t,i,k}. \qquad (19)$$

Thus PEM is a weighted superposition of top-$k$ membership probabilities; for logarithmic weights, the largest adjacent drops occur at the very top of the ranking rather than only at the truncation boundary.

Let the one-step mean increment under a single noisy ranking be $\Delta m_t(y) := \eta_m \sum_{i=1}^{\lambda} w(r_{t,i})(x_{t,i} - m_t)$, so that (30) becomes $m_{t+1} = m_t + \Delta m_t(y)$. PEM replaces the stochastic weights $w(r_{t,i})$ by $w_{t,i}^\star$:

$$\Delta m_{\text{PEM},t} := \eta_m \sum_{i=1}^{\lambda} w_{t,i}^\star (x_{t,i} - m_t). \tag{20}$$

This update is deterministic for given $x_{t,1:\lambda}$ and integrates over ranking uncertainty.

**Lemma 1.** *For the standard rank-based mean increment $\Delta m_t(y)$ defined above,*

$$\mathbb{E}[\Delta m_t(y) \mid x_{t,1:\lambda}] = \Delta m_{\text{PEM},t}. \tag{21}$$

*Proof sketch.* Conditioning on $x_{t,1:\lambda}$ makes $(x_{t,i} - m_t)$ deterministic; then taking conditional expectations linearly replaces $w(r_{t,i})$ by $\mathbb{E}[w(r_{t,i}) \mid x_{t,1:\lambda}] = w_{t,i}^\star$. $\qquad\square$

CMA-ES state updates depend on $y_{t,1:\lambda}$ only through rank-weighted statistics ((32)–(34)). In our implementation, PEM is applied by replacing each occurrence of $w(r_{t,i})$ with an estimate of $w_{t,i}^\star$.

### 4.2. Residual Bootstrapped PEM (RB-PEM)

Computing PEM $w_{t,i}^\star$ exactly would require extensive reevaluations per candidate, risking depth collapse (13). Residual bootstrapping estimates $w_{t,i}^\star$ with a capped reevaluation budget per generation as (16), in which $K_t \leq K_{\max} \ll \lambda$ counts only additional reevaluations used to calibrate the bootstrap noise model. As shown by (17), this keeps the depth close to $B/\lambda$ whenever $\mathbb{E}[K_t] \ll \lambda$.

In Bayesian context, $w_{t,i}^\star$ is a posterior expectation of a rank weight under the local evaluation-noise law. A parametric Bayesian alternative would sample latent values from a fitted likelihood/posterior, rank them, and average $w(\cdot)$ over those ranks. RB-PEM uses the same Monte Carlo structure, but draws synthetic values from a nonparametric residual pool. This avoids specifying a likelihood for non-Gaussian, heteroscedastic, or state-dependent noise, amortizes noise information across generations, and uses the same bootstrap rankings for any choice of $w$.

In practice, residual bootstrapping combines two complementary design principles aligned with the fixed-budget protocol, see Appendix B for details of implementation. Note that residual bootstrapping could have statistical risk of *residual-pool mismatch* due to finite pool size, nonstationary drift, covariate-dependent shape changes, or misspecified center/scale standardization. A convenient way to quantify mismatch is the Wasserstein-1 distance $W_1(\widehat{D}_t, D_t)$ between the pool and a target distribution $D_t$. Detailed diagnostics and mismatch decompositions are deferred to Appendix C. Empirical tests can be found in Appendix F.

### 4.3. Probe-and-Switch

Residual bootstrapping is designed for *high-misranking* regimes. When the ranking is already stable, smoothing can weaken selection pressure and the reevaluation budget $K_t$ becomes overhead. We therefore use a low-cost probe to decide whether to run RB-PEM or to run standard CMA-ES. See Appendix D for details.

## 5. Theory

This section sets a theoretical foundation for our method, which clearly shows the following: (i) PEM can be viewed as a *Rao–Blackwellization* Lehmann & Casella (1998); Casella & Berger (2002) of the standard noisy rank-based update; (ii) under *local curvature*, conditional update dispersion incurs an inevitable expected objective loss; (iii) RB-PEM is effective whenever the residual pool distribution is close (in $W_1$) to the local standardized noise distribution and *near ties* are rare.

### 5.1. PEM as the Conditional Mean Update

Lemma 1 already states the key identity: $\Delta m_{\text{PEM}} = \mathbb{E}[\Delta m(y) \mid x_{1:\lambda}]$. This identity can be simply viewed as a Rao–Blackwellization with respect to evaluation noise: conditioning on the candidate set integrates out ranking randomness without changing the conditional mean update.

**Lemma 2.** *Let $\Delta m(y) \in \mathbb{R}^d$ be any square-integrable random update and let $\mathcal{G} := \sigma(x_{1:\lambda})$ be the $\sigma$-field generated by the sampled candidates. Then among all $\mathcal{G}$-measurable (candidate-deterministic) vectors $a(x_{1:\lambda})$,*

$$\Delta m_{\text{PEM}} = \mathbb{E}[\Delta m(y) \mid \mathcal{G}]$$
$$= \arg \min_{a \in \mathcal{G}\text{-measurable}} \mathbb{E}\big[\|\Delta m(y) - a\|^2 \mid \mathcal{G}\big]. \tag{22}$$

### 5.2. Update Dispersion Induces Curvature Loss

The next result quantifies how evaluation noise manifests itself as *curvature loss*: even when the conditional mean update is unchanged, a dispersed (random) update incurs extra expected objective value in a locally curved region.

**Theorem 1.** *Let $X = m + \Delta m(y)$ and $\bar{X} = \mathbb{E}[X \mid x_{1:\lambda}] = m + \Delta m_{\text{PEM}}$. Then under Assumption 1 (Appendix C),*

$$\mathbb{E}[f(X) \mid x_{1:\lambda}] \geq f(\bar{X}) + \frac{\alpha}{2} \mathbb{E}\big[\|X - \bar{X}\|^2 \mid x_{1:\lambda}\big]. \tag{23}$$

Theorem 1 shows that, under the strong-convexity Assumption 1, dispersion incurs a Jensen gap (see the proof in Appendix E). Note that for $\alpha = 0$ (mere convexity), Theorem 1 reduces to the conditional Jensen inequality $\mathbb{E}[f(X) \mid x_{1:\lambda}] \geq f(\bar{X})$. Equation (23) says that even if a noisy rank-based strategy has the same conditional mean update as PEM, its conditional dispersion still creates an

inevitable *expected objective penalty*. This turns "ranking noise" into an explicit per-generation loss term proportional to $\mathbb{E}[\|X - \bar{X}\|^2 \mid x_{1:\lambda}]$.

### 5.3. Fixed-Budget Prediction: Per-Evaluation Dispersion Matters

We now translate Theorem 1 into a fixed-budget comparison. The key point is that, under the fixed-budget protocol, dispersion per evaluation outweighs dispersion per generation. Theorem 1 shows that, on a region where $f$ is locally $\alpha$-strongly convex, conditional update dispersion produces a local expected objective penalty of at least $\frac{\alpha}{2}\mathbb{E}[\|X - \bar{X}\|^2 \mid x_{1:\lambda}]$ in that generation. Proposition 1 in Appendix C shows that any additional reevaluation overhead reduces depth $T$ under a fixed budget. Consequently, in strictly fixed-budget regimes, the relevant efficiency notion is how much conditional dispersion is reduced *per additional oracle call*. Our approach targets dispersion reduction by integrating ranking uncertainty at the selection stage via PEM, while keeping the reevaluation overhead $K_t$ capped and reusable.

### 5.4. Residual Bootstrapping Approximation of PEM

Residual bootstrapping approximates $w_i^\star = \mathbb{E}[w(r_i) \mid x_{1:\lambda}]$ by simulating many pseudo-rankings from a fitted noise model. This subsection makes explicit when this approximation is accurate.

Let $w : \{1, \ldots, \lambda\} \to \mathbb{R}$ be the deterministic rank-weight map, and define $\|w\|_\infty := \max_k |w(k)|$ and $\Delta_w := \max_k |w(k+1) - w(k)|$. For a fixed candidate set $\{x_{1:\lambda}\}$, define the "true" expected rank weight under $D_t$ similarly with (18) as

$$w_i^\star = \mathbb{E}[w(\text{rank}_i(y_{1:\lambda}))|x_{1:\lambda}],$$

and define the analogous bootstrap expectation

$$\widetilde{w}_i = \mathbb{E}[w(\text{rank}_i(\tilde{y}_{1:\lambda}))|x_{1:\lambda}], \quad \tilde{y}_i := f(x_i) + s(x_i)\hat{\varepsilon}_i.$$

The key difficulty is that ranks are discontinuous functions of the noise vector. In Appendix C, under reasonable assumptions (Assumption 2 and 3), Proposition 2 reduces RB-PEM inaccuracy to the Wasserstein mismatch $W_1(\widehat{D}_t, D_t)$. Then Proposition 3 shows how this mismatch decomposes into a finite-pool term plus three bias terms. Combined with Proposition 4, this yields a concrete checkable route from residual-pool mismatch to expected-weight error. Together with Theorem 1, this supports the depth-over-fidelity mechanism: under fixed budgets, a small additive overhead $K_t$ is worthwhile precisely when it reduces conditional update dispersion enough to offset its cost.

### 5.5. Decision Theory for Probe-and-Switch

Define the conditional advantage of RB-PEM over CMA-ES given probe value $p$ as

$$\Delta(p) \triangleq \mathbb{E}[L_0 - L_1 \mid P = p]. \tag{24}$$

Thus, $\Delta(p) > 0$ means RB-PEM has lower conditional expected loss. Proposition 5 of Appendix D indicates that wherever $\Delta(p)$ has sign alternating provides a potential switch threshold.

## 6. Experiments

### 6.1. Experimental Setup

All comparisons follow the fixed $B$ budget accounting of (3). Performance is measured by the simple regret (4) of the final recommendation $\hat{x}_B$. We report $\log_{10}(f(\hat{x}_B) - f^\star)$ throughout, where smaller values indicate better performance.

Our primary benchmark is the `bbob-noisy` suite from COCO, comprising 30 functions $\times$ 15 instances $\times$ 3 dimensions ($d \in \{10, 20, 40\}$), yielding 450 problem instances per dimension. Unless otherwise stated, we use the default CMA-ES population size $\lambda = 4 + \lfloor 3 \ln d \rfloor$ (giving $\lambda{=}15$, $\mu{=}7$ at $d{=}40$) and identical initialization across methods. RB-PEM uses bootstrap size $B_{\text{boot}}{=}32$ and reevaluation cap $K_{\max}{=}1$ throughout. Table 1 reports aggregate win/loss counts on all 30 functions at $B = 200d$; following the probe-statistic clustering rule (Appendix F.12), Figures 1–2 and the appendix ablations focus on a high-misranking subset of 15 functions at $d = 40$ with $B = 100d$, where misranking dominates.

Baselines include: (i) vanilla CMA-ES; (ii) fixed-$k$ resampling (Res.($k$)) with $k \in \{5, 10\}$; (iii) UH-CMA-ES; and (iv) RB-PEM. There are six external tasks with 50 random seeds each: LQR control, Breast Cancer classification, Digits recognition, CartPole-HT (heavy-tailed), standard CartPole, and Pendulum (Table 1).

**Reproducibility**. We provide one-click reproducible source code at `https://github.com/sichen-wang/Depth-over-Fidelity_ICML2026`.

### 6.2. Main Results on COCO bbob-noisy

Figure 1 presents fixed-budget convergence curves on four representative `bbob-noisy` functions, showing that RB-PEM achieves substantially lower final regret than evaluation-stage denoising baselines while maintaining near-maximal depth. A key observation is that depth markers are almost non-overlapping across methods: in these high-misranking regimes, greater depth strongly correlates with better final performance, supporting our depth-over-fidelity thesis.

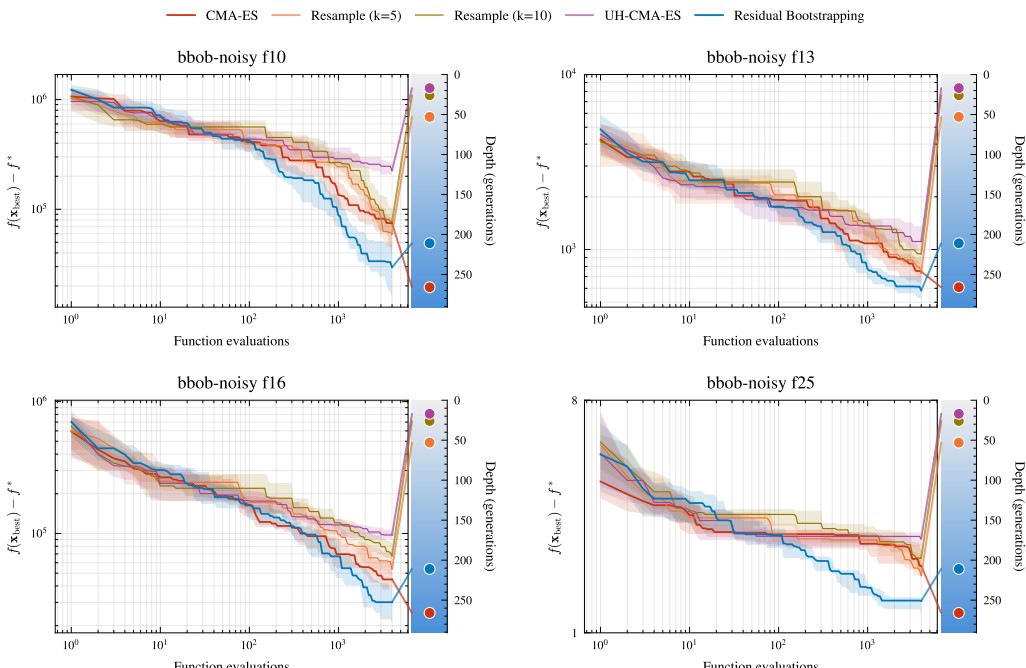

*Figure 1.* **RB-PEM improves fixed-budget progress on COCO `bbob-noisy`.** Median (solid) and interquartile range (shaded) of $\log_{10}(f(\hat{x}) - f^\star)$ versus evaluations on four representative high-misranking functions (f110, f113, f116, f125; $d = 40$, $B = 100d$, 15 instances each). Right-side markers indicate the number of completed CMA-ES generations (depth). Methods that preserve depth (CMA-ES, RB-PEM) generally achieve lower final regret; among these, RB-PEM further improves by integrating ranking uncertainty at the selection stage rather than spending evaluations on per-candidate denoising. Evaluation-stage baselines (Resample, UH-CMA-ES) sacrifice depth for per-generation fidelity, resulting in higher final regret despite cleaner intra-generation rankings. *Protocol:* Each of the 15 COCO instances is run once per method (no repeated seeds); lines show the median and shading spans the 25th–75th percentile (IQR) across instances.

## 6.3. Depth–Fidelity Trade-off

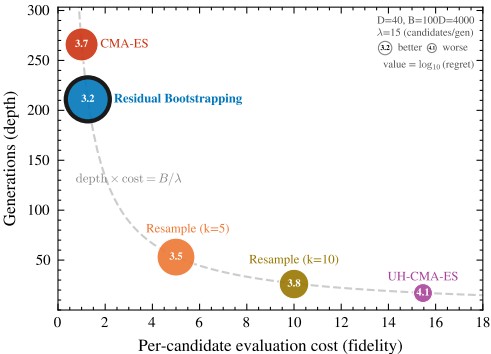

*Figure 2.* **Depth–fidelity trade-off under a fixed budget.** Each method is plotted by its average per-candidate evaluation cost (fidelity, $x$-axis) and number of completed generations (depth, $y$-axis) on the high-misranking COCO `bbob-noisy` subset ($d = 40$, $B = 100d$, 225 problems = 15 functions × 15 instances). Bubble annotations encode the median final $\log_{10}$ regret (smaller is better). The grey hyperbola marks the budget constraint depth × cost ≈ $B/\lambda$: higher fidelity necessarily reduces depth. RB-PEM achieves the lowest regret while remaining in the low-cost / high-depth region, supporting the thesis that selection-stage uncertainty integration is more sample-efficient than evaluation-stage denoising under strict budgets. *Protocol:* Each instance is run once per method; each bubble represents one method and its annotation reports the median final regret across all 225 instances.

Figure 2 confirms the depth–fidelity trade-off of Section 3.4 empirically: under a fixed budget the baselines spread along the depth–fidelity frontier, and RB-PEM occupies the low-cost/high-depth corner that minimizes regret.

## 6.4. Probe-and-Switch Evaluation

Figure 3 shows RB-PEM excelling under severe ranking noise (e.g., heavy-tailed RL rollouts, noisy HPO) but slipping below vanilla CMA-ES on low-misranking tasks, where its extra smoothing and reevaluations become unnecessary overhead—motivating an adaptive policy.

We verify that the high-misranking subset used in Figures 1–3 is not a post-hoc selection: Figure 4 sorts all 30 `bbob-noisy` functions by the probe statistic $P$ and reveals a sharply bimodal structure (gap 0.145, ≈ 3× either cluster's width) that yields the same partition for any $\tau_{\text{cluster}} \in [0.16, 0.29]$. The probe also predicts *when* uncertainty integration helps: RB-PEM and Probe-and-Switch lead on the high-misranking cluster but stay within roughly one rank of the others on the low-misranking cluster.

Figure 5 plots the conditional advantage $\Delta(p) = \mathbb{E}[L_0 - L_1 \mid P = p]$ against the probe statistic $P$. The curve exhibits approximate *single crossing*—$\Delta(p) < 0$ for small

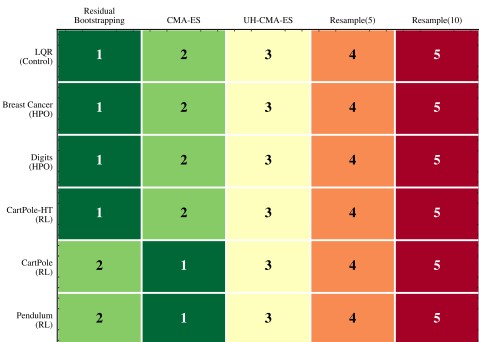

*Figure 3.* **Task-level algorithm ranking** (1 = best). Median rank across instances for each method (columns) on each task (rows): 15 high-misranking COCO `bbob-noisy` functions and six external tasks ($d = 40$, $B = 100d$ for COCO; task-specific budgets otherwise; darker is better). RB-PEM ranks first on most high-misranking tasks but underperforms CMA-ES on low-misranking tasks (e.g., CartPole, Pendulum), where the bootstrap overhead outweighs the smoothing benefit. This pattern motivates probe-and-switch (Section 5.5). *Protocol:* 1 run per COCO instance (15 each), 50 seeds per external task (Appendix: per-task budgets/noise); ranks use each task's median objective.

$P$ (stable ranks) and $\Delta(p) > 0$ beyond a threshold—which enables a **probe-and-switch** policy: route tasks with $P \geq \tau$ ($\tau = 0.12$) to RB-PEM and the rest to CMA-ES, yielding purely positive improvement under the model of Section 5.5.

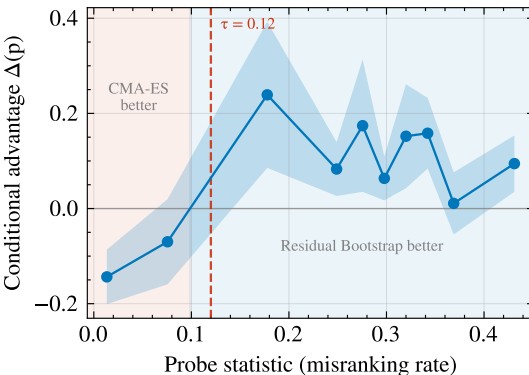

*Figure 5.* **Conditional advantage and single crossing.** Estimated $\Delta(p) = \mathbb{E}[L_{\text{CMA}} - L_{\text{RB-PEM}} \mid P = p]$ vs. the probe statistic $P$ (normalized rank disagreement, Eq. (56)) on all 30 COCO `bbob-noisy` functions ($d = 40$, $B = 500d$); positive $\Delta$ favors RB-PEM. The curve crosses zero once near $\tau = 0.12$ (dashed line), giving a threshold rule (RB-PEM when $P \geq \tau$, else CMA-ES) that is Bayes-optimal under single crossing (Proposition 5). *Protocol:* 450 problems (30 functions × 15 instances), 1 run each; bin means over 12 quantile bins, error bars ±1.96 SE (95% CI).

Scanning thresholds on COCO identifies two robust operating points: an aggressive $\tau = 0.12$ (switch more often) and a conservative $\tau = 0.22$ (switch only under strong misranking evidence). Figure 6 shows these COCO-calibrated thresholds transfer reasonably to external tasks, though new domains may warrant re-tuning.

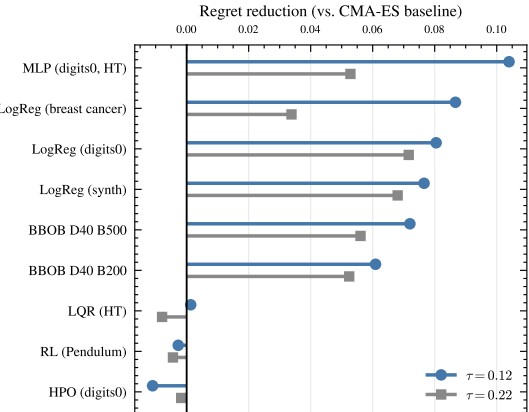

*Figure 6.* **Threshold transfer to external tasks.** Regret reduction of Probe-and-Switch relative to CMA-ES on nine transfer targets (COCO and external), using COCO-calibrated thresholds $\tau \in \{0.12, 0.22\}$ without per-task re-tuning. The aggressive $\tau = 0.12$ activates RB-PEM more often, yielding larger reductions on high-misranking tasks (e.g., LQR, MLP), while the conservative $\tau = 0.22$ mitigates negative transfer on low-misranking ones (e.g., HPO, Pendulum). *Protocol:* mean regret reduction per target (over instances/seeds), ordered by $\Delta(\tau=0.12)$; 1 run per COCO instance, 50 seeds per external task.

### 6.5. Comprehensive Comparison

Table 1 reports pairwise win/loss counts for probe-and-switch against each competitor across COCO and six external tasks. It wins decisively against all evaluation-stage baselines (Res.(10), Res.(5), UH-CMA-ES; all above 77%) and holds a moderate advantage over its constituent algorithms (CMA-ES, RB-PEM), indicating that the switch reliably activates the beneficial mode without unnecessary overhead. Its significance against CMA-ES strengthens with dimension, consistent with misranking becoming more problematic in higher dimensions; on external tasks it tracks RB-PEM under high rank instability and defaults to CMA-ES otherwise.

Appendix F provides controlled mechanism checks: (i) tests linking higher misranking to larger update dispersion and objective loss, and (ii) ablations confirming RB-PEM's persistent gain under tightly capped reevaluation, supporting the "selection-stage integration" interpretation.

## 7. Conclusion

In strictly fixed-budget noisy black-box optimization, the cost of improving intra-generation ranking fidelity is paid in reduced *depth*: every additional evaluation spent on denoising or resampling shortens the number of distribution updates that can be executed before the budget is exhausted. By making this accounting explicit, we formalized a depth–fidelity trade-off for noisy rank-based evolution strategies and argued that, in high-misranking regimes, the lost depth can dominate the gains from per-generation fidelity.

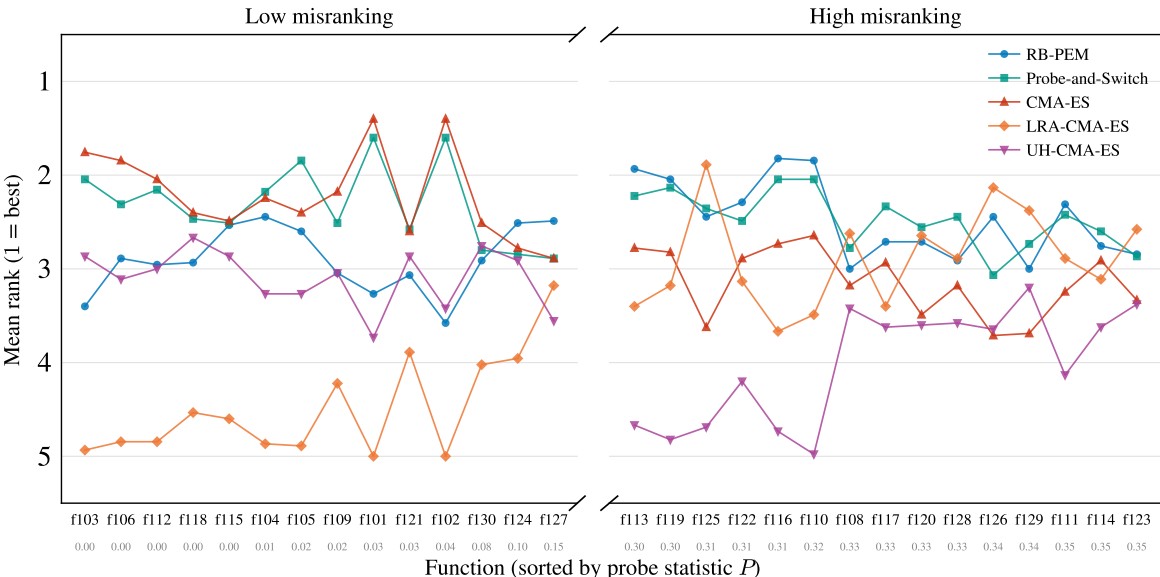

*Figure 4.* **Per-function ranking across all 30 `bbob-noisy` functions reveals a bimodal misranking structure that defines the high-misranking subset.** Mean rank (1 = best) of five methods, sorted by the median probe statistic $P$ (Eq. (56)). The $P$-values are sharply bimodal: 14 *low-misranking* functions cluster in $P \in [0.00, 0.15]$ (left) and 15 *high-misranking* ones in $P \in [0.30, 0.35]$ (right), separated by a gap of 0.145 ($\approx 3\times$ either cluster's width), so any threshold $\tau_{\text{cluster}} \in [0.16, 0.29]$ produces the same partition. RB-PEM and Probe-and-Switch rank first or second on the high-misranking cluster, while LRA-CMA-ES and UH-CMA-ES trail; on the low-misranking cluster all five methods are within roughly one rank. We exclude f107 (high $P$ from step-ellipsoidal structure, not noise); the remaining 15 functions are the subset used in Figures 1–3 and the appendix ablations. *Protocol:* $B$=100$d$; mean rank averaged over $d \in \{10, 20, 40\} \times 15$ instances per function. Probe $P$ computed at $d$=40 from 20 ES-sampled candidate sets with $(\lambda, \mu) = (32, 8)$.

| | vs Res.(10) | vs Res.(5) | vs UH-CMA-ES | vs CMA-ES | vs RB-PEM |
|---|---|---|---|---|---|
| *COCO benchmark (30 functions $\times$ 15 instances per dim, $B = 200d$)* | | | | | |
| $d = 10$ | **390/60**[***] | **364/86**[***] | **359/91**[***] | **230/210** | **281/169**[***] |
| $d = 20$ | **373/77**[***] | **356/94**[***] | **368/82**[***] | **239/195**[*] | **269/181**[***] |
| $d = 40$ | **363/87**[***] | **347/103**[***] | **359/91**[***] | **257/170**[**] | **270/180**[***] |
| *External tasks (50 seeds each)* | | | | | |
| LQR | **50/0**[***] | **46/4**[***] | **46/4**[***] | **31/19** | 23/27 |
| Breast Cancer | **50/0**[***] | **37/13**[***] | 25/25 | 20/29 | 22/28 |
| Digits | **49/1**[***] | **40/10**[***] | **31/19** | 17/33[*] | 17/33[*] |
| CartPole-HT | **48/2**[***] | **43/6**[***] | **30/19** | 21/23 | 20/24 |
| CartPole | **44/6**[***] | **36/14**[**] | **26/24** | 19/30 | 21/27 |
| Pendulum | **46/4**[***] | **38/11**[***] | **32/18** | **31/19** | **31/19** |
| **Total W/L** | **1413/237** | **1307/341** | **1276/373** | **865/728** | **954/688** |
| **Win Rate** | **85.6%** | **79.3%** | **77.4%** | **54.3%** | **58.1%** |

*Table 1.* **Probe-and-Switch vs. competitors**: pairwise win/loss (W/L) counts on COCO `bbob-noisy` (30 functions $\times$ 15 instances per dim, $B = 200d$, $d \in \{10, 20, 40\}$) and six external tasks (50 seeds each). Bold indicates Probe-and-Switch wins ($W > L$). Stars denote two-sided Wilcoxon signed-rank significance: [***]$p < 0.001$, [**]$p < 0.01$, [*]$p < 0.05$. Res.($k$): $k$-fold fixed resampling; RB-PEM: residual-bootstrap PEM without probe-and-switch. *Protocol:* Probe-and-Switch uses threshold $\tau$=0.12; each COCO instance is run once per method and each external task with 50 independent seeds.

We operationalized *selection-stage uncertainty integration* as *probabilistic elite membership* (PEM)—a Rao–Blackwellization of the noisy rank-based update that replaces hard rank assignments with expected rank weights, preserving the mean update while reducing its disper-

sion. We approximated PEM at near-unit cost via *residual-bootstrapped PEM* (RB-PEM), which calibrates a local noise model from a small capped reevaluation set, amortized across generations via pooled residuals and backed by a falsifiable mismatch decomposition and runtime diagnostics; a low-cost *probe-and-switch* policy reverts to vanilla CMA-ES when ranks are stable.

Our theory clarifies why depth-preserving uncertainty integration can win: under local curvature, conditional update dispersion induces an unavoidable expected objective penalty, making *dispersion reduction per oracle call* the relevant efficiency metric in fixed-budget regimes. Empirically, RB-PEM achieves consistently steeper fixed-budget progress on the COCO bbob-noisy suite and diverse external tasks, outperforming evaluation-stage denoising baselines whose higher per-generation costs collapse depth, while probe-and-switch improves robustness across regimes.

Overall, the results support a testable thesis: *when budgets are strict and ranking uncertainty is high, integrating uncertainty into selection is more sample-efficient than spending evaluations to eliminate it*. Promising directions include extending the residual model to correlated and nonstationary noise, and porting depth-over-fidelity to other population-based optimizers.

## Impact Statement

RB-PEM enables noisy black-box optimizers to extract more progress from a fixed number of evaluations, which directly reduces the computational cost of tasks such as RL policy search with stochastic rollouts and hyperparameter optimization with noisy validation. As a general-purpose algorithmic improvement, it inherits the dual-use profile common to all black-box optimizers; we are not aware of any application-specific risks particular to our method.

## Acknowledgments

The first author would like to thank his parents for their unwavering support and encouragement; his algorithm competition coaches Wenwu Wang and Yuan Sun, who sparked his interest in computer science; his ICPC teammates Zhang Chen, An Yan, and Yuqi Peng, and the many fellow competitors who accompanied him through eight years of algorithmic contests; Prof. Xiaoying Tang and the members of T-Lab at CUHK-Shenzhen, who patiently guided a newcomer into academic research; and Siyuan Xu, whose support and companionship sustained him throughout.

This work is supported by Guangdong Province (No. 2023QN10X215), 2023 Shenzhen National Science Foundation (No. 20231128220938001), Shenzhen Science and Technology Program (No. JCYJ20241202130548062), the Natural Science Foundation of Shenzhen (No. JCYJ20230807142703006), and the Key Research Platforms and Projects of the Guangdong Provincial Department of Education (No.2023ZDZX1034).

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

## A. CMA-ES Specification

Covariance matrix adaptation evolution strategy (CMA-ES) maintains a Gaussian search distribution

$$x \sim \mathcal{N}\big(m_t, \ \sigma_t^2 C_t\big), \tag{25}$$

parameterized by mean $m_t \in \mathbb{R}^d$, global step-size $\sigma_t > 0$, and covariance $C_t \in \mathbb{R}^{d \times d}$. More explicitly, it samples $\lambda$ candidates by drawing standardized steps $z_{t,i}$ and mapping them through a factor $A_t$ of $C_t$ as follows:

$$z_{t,i} \sim \mathcal{N}(0, I_d), \quad i = 1, \ldots, \lambda, \tag{26}$$

$$x_{t,i} = m_t + \sigma_t A_t z_{t,i}, \quad A_t A_t^\top = C_t. \tag{27}$$

The mean $m_t$ is updated according to the reordering of observed values. Let $\pi_t \in S_\lambda$ be a permutation such that

$$y_{t,\pi_t(1)} \le y_{t,\pi_t(2)} \le \cdots \le y_{t,\pi_t(\lambda)}, \tag{28}$$

with ties broken deterministically (e.g., by index). The rank of individual $i$ is $r_{t,i} \in \{1, \ldots, \lambda\}$ defined by $\pi_t(r_{t,i}) = i$. Equivalently, for each candidate $i \in \{1, \ldots, \lambda\}$ we define its rank by

$$r_{t,i} := 1 + \sum_{j=1}^{\lambda} \mathbf{1}\{ y_{t,j} < y_{t,i} \} + \sum_{j=1}^{\lambda} \mathbf{1}\{ y_{t,j} = y_{t,i}, j < i \}, \tag{29}$$

so that $r_{t,i} \in \{1, \ldots, \lambda\}$ and $\pi_t(r_{t,i}) = i$ (ties are broken by index). Then the mean is updated by

$$m_{t+1} = m_t + \eta_m \sum_{i=1}^{\lambda} w(r_{t,i})(x_{t,i} - m_t), \tag{30}$$

where $w : \{1, \ldots, \lambda\} \to \mathbb{R}$ is a deterministic rank-weight map. For the mean update in standard positive-weight CMA-ES, these weights are nonnegative and normalized, $\sum_{j=1}^{\lambda} w(j) = 1$. The default choice in modern CMA-ES implementations is the positive logarithmic recombination weight

$$w(j) = \frac{\max\{\log(\mu + 1/2) - \log j, \ 0\}}{\sum_{\ell=1}^{\mu} \big( \log(\mu + 1/2) - \log \ell \big)}, \qquad j = 1, \ldots, \lambda. \tag{31}$$

Uniform top-$\mu$ truncation, $w(j) = \frac{1}{\mu}\mathbf{1}\{j \le \mu\}$, is an older/intermediate-recombination special case and is used in this paper only when explicitly stated as a pedagogical example. Finally, the remaining state variables $(\sigma_t, C_t)$ are updated using the standard cumulative step-size and covariance adaptation rules (Hansen & Ostermeier, 2001).

Define the rank-weighted step in standardized coordinates

$$z_{w,t} := \sum_{i=1}^{\lambda} w(r_{t,i})\, z_{t,i} = \sum_{j=1}^{\lambda} w(j)\, z_{t,\pi_t(j)}. \tag{32}$$

The corresponding recombination point is

$$x_{w,t} := \sum_{j=1}^{\lambda} w(j)\, x_{t,\pi_t(j)} = m_t + \sigma_t A_t z_{w,t}. \tag{33}$$

Thus the mean update (30) can be written equivalently as

$$m_{t+1} = m_t + \eta_m\,(x_{w,t} - m_t) = m_t + \eta_m\, \sigma_t A_t z_{w,t}. \tag{34}$$

Crucially, the update depends on the observed values only through their *relative ordering* (28); under noisy evaluations, the ranks $r_{t,i}$ (and hence $z_{w,t}$) are random even conditional on the queried points.

## B. Residual Bootstrapping Implementation Details

Implementing residual bootstrapping follows:

**Step 1: targeted reevaluation and robust residuals**. After one baseline evaluation $y_{t,i}$ for each candidate, we form the observed ranks $\hat{r}_{t,i}$ (Section 3.3) and select a small reevaluation set $\mathcal{B}_t$ subject to the cap $K_t \leq K_{\max}$. For uniform top-$\mu$ weights, the weight-instability criterion of Appendix C reduces to a narrow boundary band, for example $\hat{r}_{t,i} \in \{\mu - 1, \mu, \mu + 1\}$. For non-uniform monotone weights, the same criterion ranks candidates by bootstrap instability of $w(\tilde{r}_{t,i})$ and can include uncertain top-ranked candidates as well. We allocate $R_{t,i}$ additional independent reevaluations to each $i \in \mathcal{B}_t$ with $\sum_{i \in \mathcal{B}_t} R_{t,i} = K_t$ and $K_t \leq K_{\max}$. Let $y_{t,i}^{(1)} := y_{t,i}$ and $y_{t,i}^{(2)}, \ldots, y_{t,i}^{(1+R_{t,i})}$ denote these samples. We compute a robust per-candidate median over reevaluated points,

$$\tilde{m}_{t,i} := \operatorname{median}_{1 \leq k \leq 1 + R_{t,i}} y_{t,i}^{(k)}, \qquad i \in \mathcal{B}_t,$$

and residuals $\epsilon_{t,i}^{(k)} := y_{t,i}^{(k)} - \tilde{m}_{t,i}$.

**Step 2: denoised centering model $\hat{f}_t(\cdot)$ (cross-fitted)**. The bootstrap values should be centered at an estimate of the latent mean $f(x)$, not at the particular noisy draw $y_{t,i}$, otherwise the bootstrap targets a noise-convolved distribution. We fit a lightweight predictor $\hat{f}_t(\cdot)$ in standardized CMA-ES coordinates $z_{t,i} = (x_{t,i} - m_t)/\sigma_t$ using the same $m_t, \sigma_t$ as in sampling (Appendix A), with *cross-fitting* so that $\hat{f}_t(x_{t,i})$ does not reuse $y_{t,i}$ when generating bootstrap values for candidate $i$. A concrete instantiation is a two-fold ridge regression on pseudo-labels that use $\tilde{m}_{t,i}$ for $i \in \mathcal{B}_t$ and $y_{t,i}$ otherwise. See full details of implementation in Appendix.

**Step 3: optional input-dependent noise scaling $\hat{s}_t(\cdot)$ and winsorization**. To accommodate state-dependent noise scales, we optionally fit a simple scale model $\hat{s}_t(x)$ using only boundary points, for example, a two-parameter linear model in $|\hat{f}_t(x)|$, and we winsorize standardized residuals at a fixed threshold $M$. If input-dependent noise is weak or unknown, one may set $\hat{s}_t(\cdot) \equiv \hat{s}_t$ constant.

**Step 4: residual pool $\widehat{D}_t$ amortized across time**. We insert winsorized standardized residuals into a pool:

$$\hat{z}_{t,i}^{(k)} := \operatorname{clip}\left(\frac{\epsilon_{t,i}^{(k)}}{\hat{s}_t(x_{t,i})}, [-M, M]\right), \qquad i \in \mathcal{B}_t,$$

and let $\widehat{D}_t$ denote the empirical distribution of all pooled residuals collected up to generation $t$. This amortization is the main reason why residual bootstrapping can run many bootstrap rankings without consuming additional calls.

**Step 5: bootstrap rankings and expected weights**. Given $\hat{f}_t(\cdot)$, $\hat{s}_t(\cdot)$, and $\widehat{D}_t$, we generate synthetic noisy values

$$\tilde{y}_{t,i}^{(b)} = \hat{f}_t(x_{t,i}) + \hat{s}_t(x_{t,i}) z_{t,i}^{(b)}, \qquad z_{t,i}^{(b)} \overset{iid}{\sim} \widehat{D}_t. \tag{35}$$

For each bootstrap replicate $b = 1, \ldots, B_{\text{boot}}$, we compute the induced ranks $\tilde{r}_{t,i}^{(b)} = \operatorname{rank}_i(\tilde{y}_{t,1:\lambda}^{(b)})$ and estimate expected rank weights by averaging:

$$\hat{w}_{t,i} = \frac{1}{B_{\text{boot}}} \sum_{b=1}^{B_{\text{boot}}} w(\tilde{r}_{t,i}^{(b)}). \tag{36}$$

It only requires sorting synthetic values, since it is free evaluation-wise once $\widehat{D}_t$ is built.

In summary, residual bootstrapping provides averaging estimates $\hat{w}_{t,i}$ of the PEM target weights $w_{t,i}^\star$ (18) via bootstrap pseudo-rankings, which are then used as a drop-in replacement for $w(r_{t,i})$ in the mean update (30) and any other rank-weighted CMA-ES statistics. Crucially, this preserves the *one-evaluation-per-candidate* baseline and spends the capped overhead $K_t$ only to calibrate the bootstrap noise model, yielding the per-generation evaluation cost $C_t = \lambda + K_t$ in (16). This design also cleanly separates *evaluation-cost knobs*, which directly affect $C_t$ ($K_{\max}$ and the boundary bandwidth defining $\mathcal{B}_t$), from *compute-only knobs* (the number of bootstrap replicates $B_{\text{boot}}$), which can be increased to stabilize $\hat{w}_{t,i}$ without changing the budget accounting in (16).

## C. Residual Pool Theory and Diagnostics

### C.1. Fixed-Budget Condition ("Money Plot" Prediction)

**Proposition 1.** *Let $B$ be the total number of allowed oracle calls and let the per-generation cost be $C_t = \lambda + K_t$ with $K_t \in [0, K_{\max}]$. Then the number of completed generations $T$ satisfies*

$$\left\lfloor \frac{B}{\lambda + K_{\max}} \right\rfloor \leq T \leq \left\lfloor \frac{B}{\lambda} \right\rfloor. \tag{37}$$

*In addition, if $(K_t)_{t \geq 0}$ is i.i.d. with $\mathbb{E}[K_t] < \infty$, then for large budgets one has the approximation*

$$\mathbb{E}[T] \approx \frac{B}{\lambda + \mathbb{E}[K_t]}. \tag{38}$$

*Proof of Proposition 1.* The bounds (37) are immediate from $\lambda \leq C_t \leq \lambda + K_{\max}$ and $\sum_{t=0}^{T-1} C_t \leq B$.

For (38), assume $(C_t)_{t \geq 0}$ are i.i.d. with $\mu := \mathbb{E}[C_0] < \infty$. Define $S_n := \sum_{t=0}^{n-1} C_t$ and the stopping time $T := \max\{n : S_n \leq B\}$. Since $C_t \geq \lambda > 0$, we have $T \leq \lfloor B/\lambda \rfloor$, hence $T$ is bounded. Let $M_n := S_n - n\mu$. Then $(M_n)$ is a martingale, so by optional stopping, $\mathbb{E}[M_T] = \mathbb{E}[M_0] = 0$ and therefore $\mathbb{E}[S_T] = \mu \mathbb{E}[T]$ (Wald's identity (Williams, 1991)). Because $S_T \leq B < S_{T+1}$, we get $\mu \mathbb{E}[T] = \mathbb{E}[S_T] \leq B$ and $B < \mathbb{E}[S_{T+1}] = \mu \mathbb{E}[T+1] = \mu(\mathbb{E}[T] + 1)$, so $B/\mu - 1 < \mathbb{E}[T] \leq B/\mu$. With $\mu = \lambda + \mathbb{E}[K_t]$ in condition, this yields $\mathbb{E}[T] = B/(\lambda + \mathbb{E}[K_t]) + O(1)$, proving (38) for large $B$. $\qquad\square$

### C.2. From Distribution Mismatch to PEM

**Assumption 1** (Localized strong convexity with localization). For any fixed candidate set $x_{1:\lambda}$, random updated point $X := m + \Delta m(y)$ and its conditional mean $\bar{X} := \mathbb{E}[X \mid x_{1:\lambda}] = m + \Delta m_{\mathrm{PEM}}$, there always exists a *convex* set $\mathcal{C} \subset \mathbb{R}^d$ such that: (i) $f$ is $\alpha$-strongly convex on $\mathcal{C}$, and (ii) $X, \bar{X} \in \mathcal{C}$ almost surely (conditional on $x_{1:\lambda}$).

Note that Assumption 1 does not require global convexity. It only requires that the random update and its conditional mean stay inside a region where $f$ has curvature. A typical special case is: $f$ is $\alpha$-strongly convex on a ball $B(m, r)$ and the update is localized by design, for example, via step-size control or explicit clipping, so that $\|X - m\| \leq r$ and $\|\bar{X} - m\| \leq r$.

**Assumption 2** (Standing noise factorization). Conditional on the candidate set and history, evaluation noise is independent across candidates and has a common standardized marginal distribution.

Concretely, we assume

$$y_i = f(x_i) + s(x_i)\,\varepsilon_i, \qquad \varepsilon_i \overset{\text{i.i.d.}}{\sim} D_t, \tag{39}$$

and the bootstrap draws satisfy $\hat{\varepsilon}_i \overset{\text{i.i.d.}}{\sim} \widehat{D}_t$. If there is strong cross-candidate correlation or if the standardized shape depends sharply on $x$, then expected rank weights depend on the joint law of $(\varepsilon_1, \ldots, \varepsilon_\lambda)$ and a univariate residual pool is insufficient.

**Assumption 3** (Anti-concentration of pairwise gaps). Conditional on $x_{1:\lambda}$, for each $i \neq j$ the true gap $G_{ij} := y_i - y_j$ admits a density $p_{ij}$ in a neighborhood of 0 with

$$\sup_{|u| \leq 1} p_{ij}(u) \leq L_{ij} < \infty. \tag{40}$$

Equivalently, for either gap $H_{ij} \in \{G_{ij}, \widetilde{G}_{ij}\}$ and all $\eta \in (0, 1]$,

$$\Pr(|H_{ij}| \leq \eta \mid x_{1:\lambda}) \leq 2\eta L_{ij}. \tag{41}$$

**Proposition 2.** *Suppose both $D_t$ and $\widehat{D}_t$ are winsorized on $[-M, M]$. Then, under Assumption 2 (39) and Assumption 3, for any smoothing scale $\eta \in (0, 1]$,*

$$|\widetilde{w}_i - w_i^\star| \leq \frac{\|w\|_\infty}{\eta} \left( \sum_{j=1}^\lambda s(x_j) \right) W_1(D_t, \widehat{D}_t) + 4\Delta_w\, \eta \sum_{j \neq i} L_{ij}. \tag{42}$$

The second term in (42) scales with the mass of pairwise gaps near 0. It is small when candidate scores are well separated (large $|f(x_i) - f(x_j)|$ relative to noise), or when the gap distribution has a moderate density bound near 0. The statement is already for a general deterministic $w$: the weight map enters only through $\|w\|_\infty$ in the smooth mismatch term and through the adjacent-drop scale $\Delta_w$ in the near-tie term.

**Specialization to standard logarithmic CMA-ES weights.** For (31), let $Z_\mu := \sum_{\ell=1}^\mu (\log(\mu + 1/2) - \log \ell)$. Then

$$\delta_k = \frac{\log(1 + 1/k)}{Z_\mu}, \qquad\qquad\qquad k = 1, \ldots, \mu - 1, \qquad\qquad (43)$$

$$\delta_\mu = \frac{\log(1 + 1/(2\mu))}{Z_\mu}, \qquad \delta_k = 0, \qquad\qquad k > \mu. \qquad\qquad (44)$$

Since $Z_\mu = \mu + O(\log \mu)$, we have

$$\|w\|_\infty = \Theta\left(\frac{\log \mu}{\mu}\right), \qquad \Delta_w = \delta_1 = \Theta\left(\frac{1}{\mu}\right). \qquad\qquad (45)$$

Substituting into Proposition 2, and writing $S_x := \sum_{j=1}^\lambda s(x_j)$ and $A_i := \sum_{j \neq i} L_{ij}$, gives

$$|\widetilde{w}_i - w_i^\star| \leq C_1 \frac{\log \mu}{\mu} \frac{S_x}{\eta} W_1(D_t, \widehat{D}_t) + C_2 \frac{\eta}{\mu} A_i \qquad\qquad (46)$$

for absolute constants $C_1, C_2 > 0$. Hence Proposition 2 does not rely on uniform truncation: logarithmic weights preserve the same two-term structure, with a mild logarithmic factor only in the Wasserstein mismatch term. Uniform top-$\mu$ weights instead have $\|w\|_\infty = \Delta_w = 1/\mu$ and concentrate all adjacent-drop mass at $k = \mu$.

**Weight-aware targeted reevaluation.** The same multi-cutoff representation gives a natural general-weight version of the boundary set in Appendix B. Conditional on the fitted bootstrap model, define

$$U_{t,i} := \text{Var}\left(w(\tilde{r}_{t,i}) \mid x_{1:\lambda}, \hat{f}_t, \hat{s}_t, \widehat{D}_t\right), \qquad\qquad (47)$$

and let $\tilde{p}_{t,i,k} := \Pr(\tilde{r}_{t,i} \leq k \mid x_{1:\lambda}, \hat{f}_t, \hat{s}_t, \widehat{D}_t)$. Since the events $\{\tilde{r}_{t,i} \leq k\}$ are nested,

$$U_{t,i} = \sum_{k=1}^{\lambda-1} \sum_{\ell=1}^{\lambda-1} \delta_k \delta_\ell \left(\tilde{p}_{t,i,\min(k,\ell)} - \tilde{p}_{t,i,k} \tilde{p}_{t,i,\ell}\right). \qquad\qquad (48)$$

For uniform top-$\mu$ truncation this reduces to $U_{t,i} = \mu^{-2} \tilde{p}_{t,i,\mu}(1 - \tilde{p}_{t,i,\mu})$, maximized near the elite boundary. For logarithmic weights, multiple $\delta_k$ are positive and largest near $k = 1$, so uncertainty among highly ranked candidates contributes directly. Selecting reevaluations by the largest $U_{t,i}$ is therefore a weight-aware extension of the boundary heuristic; it recovers the original boundary rule in the one-cutoff case and expands the set toward top-ranked candidates when logarithmic weights are used.

*Proof of Proposition 2.* Step 1 (smoothing). Let $\xi \in \mathbb{R}^\lambda$ have i.i.d. coordinates $\xi_k \sim \text{Unif}[-\eta/2, \eta/2]$. Define the smoothed functional $g_{i,\eta}(u) := \mathbb{E}_\xi[w(\text{rank}_i(u + \xi))]$ for $u \in \mathbb{R}^\lambda$. This is the average of the bounded function $u \mapsto w(\text{rank}_i(u))$ over an $\ell_\infty$ cube of side length $\eta$. Therefore $g_{i,\eta}$ is $(\|w\|_\infty/\eta)$-Lipschitz w.r.t. $\|\cdot\|_1$: shifting the cube by $\delta$ changes at most a $\|\delta\|_1/\eta$ fraction of its volume, so the cube-averaged value can change by at most $\|w\|_\infty \|\delta\|_1/\eta$.

Step 2 (Wasserstein control of the smoothed mismatch). Let $Y := (y_1, \ldots, y_\lambda)$ and $\widetilde{Y} := (\tilde{y}_1, \ldots, \tilde{y}_\lambda)$. By Kantorovich–Rubinstein duality (Villani, 2008),

$$\left|\mathbb{E}[g_{i,\eta}(Y) \mid x_{1:\lambda}] - \mathbb{E}[g_{i,\eta}(\widetilde{Y}) \mid x_{1:\lambda}]\right| \leq \text{Lip}(g_{i,\eta}) \cdot W_1(\mathcal{L}(Y \mid x_{1:\lambda}), \mathcal{L}(\widetilde{Y} \mid x_{1:\lambda})),$$

where $W_1$ on $\mathbb{R}^\lambda$ uses cost $\|u - v\|_1$. Under (39), couple each $(\varepsilon_i, \hat{\varepsilon}_i)$ using an optimal 1D coupling achieving $W_1(D_t, \widehat{D}_t)$ and couple coordinates independently. Then

$$\mathbb{E}\left[\|Y - \widetilde{Y}\|_1 \mid x_{1:\lambda}\right] = \sum_{j=1}^\lambda s(x_j) \mathbb{E}[|\varepsilon_j - \hat{\varepsilon}_j|] = \left(\sum_{j=1}^\lambda s(x_j)\right) W_1(D_t, \widehat{D}_t),$$

so $W_1(\mathcal{L}(Y \mid x_{1:\lambda}), \mathcal{L}(\widetilde{Y} \mid x_{1:\lambda}))$ is bounded by the same quantity.

Step 3 (smoothing bias is a near-tie term). If $|G_{ij}| > \eta$ for all $j \neq i$, then adding $\xi \in [-\eta/2, \eta/2]^\lambda$ cannot change the sign of any pairwise gap involving $i$, hence it cannot change $\mathrm{rank}_i$. Therefore $w(\mathrm{rank}_i(Y)) \neq w(\mathrm{rank}_i(Y + \xi))$ is only possible if $|G_{ij}| \leq \eta$ for some $j \neq i$. Moreover, each adjacent rank crossing changes $w(\mathrm{rank}_i)$ by at most $\Delta_w$; summing over possible crossings involving $i$ gives a bound that is independent of the particular shape of $w$. Thus,

$$\big| w(\mathrm{rank}_i(Y)) - g_{i,\eta}(Y) \big| \leq \Delta_w \sum_{j \neq i} \mathbf{1}\{|G_{ij}| \leq \eta\},$$

and likewise for $\widetilde{Y}$. Taking expectations and applying (41) yields the second term in (42). $\qquad\square$

## C.3. Adaptive Residual Pool Concentration

**Proposition 3.** *Let $\{\widehat{z}_n\}_{n=1}^{N_t}$ be the standardized residuals stored in the pool up to time $t$, with $|\widehat{z}_n| \leq M$ winsorized. Let $\mathcal{F}_n$ denote the filtration revealing the history up to the creation of $\widehat{z}_n$. Assume conditional independence in the sense that $\widehat{z}_n$ is independent of $\{\widehat{z}_m : m < n\}$ given $\mathcal{F}_{n-1}$. Define the* path-average *distribution $\overline{D}_t := \frac{1}{N_t} \sum_{n=1}^{N_t} \mathcal{L}(\widehat{z}_n \mid \mathcal{F}_{n-1})$ and the empirical pool distribution $\widehat{D}_t := \frac{1}{N_t} \sum_{n=1}^{N_t} \delta_{\widehat{z}_n}$. Then there exists a constant $C > 0$ such that, for any $\delta \in (0, 1)$,*

$$W_1(\widehat{D}_t, \overline{D}_t) \leq C \, M \, \sqrt{\frac{\log(1/\delta)}{N_t}} \tag{49}$$

*with probability at least $1 - \delta$.*

*Proof of Proposition 3.* We use the Kantorovich–Rubinstein dual representation (Villani, 2008):

$$W_1(\widehat{D}_t, \overline{D}_t) = \sup_{\mathrm{Lip}(\varphi) \leq 1} \left| \frac{1}{N_t} \sum_{n=1}^{N_t} \varphi(\widehat{z}_n) - \frac{1}{N_t} \sum_{n=1}^{N_t} \mathbb{E}[\varphi(\widehat{z}_n) \mid \mathcal{F}_{n-1}] \right|.$$

Fix a 1-Lipschitz $\varphi$ with $\varphi(0) = 0$. Then $|\varphi(\widehat{z}_n)| \leq M$ and

$$d_n := \varphi(\widehat{z}_n) - \mathbb{E}[\varphi(\widehat{z}_n) \mid \mathcal{F}_{n-1}]$$

is a bounded martingale difference with $|d_n| \leq 2M$. Azuma–Hoeffding ((Azuma, 1967), (Hoeffding, 1963)) therefore yields, for any $t > 0$,

$$\Pr\left( \left| \frac{1}{N_t} \sum_{n=1}^{N_t} d_n \right| \geq t \right) \leq 2 \exp\left( -\frac{N_t t^2}{8M^2} \right).$$

To pass from a fixed $\varphi$ to the supremum over all 1-Lipschitz functions, cover the unit-Lipschitz class on $[-M, M]$ by an $\varepsilon$-net in $\| \cdot \|_\infty$ (piecewise-linear interpolation on a uniform grid suffices) and apply a union bound; see, e.g., van der Vaart & Wellner (1996); Boucheron et al. (2013). Choosing $\varepsilon$ on the order of $1/\sqrt{N_t}$ gives (49). $\qquad\square$

## C.4. A Drift-Aware Mismatch Decomposition (Four Observable Terms)

**Proposition 4.** *Assume winsorization at level $M$. Let $\widehat{D}_t$ and $\overline{D}_t$ be the empirical pool and path-average distributions from Proposition 3. For each stored residual index $n \in \{1, \ldots, N_t\}$, let $\tau_n$ denote the generation at which $\widehat{z}_n$ was created (so $\tau_n$ is $\mathcal{F}_{n-1}$-measurable), and define the corresponding* ideal *(oracle) standardized residual*

$$z_n^\star := \mathrm{clip}\left( \frac{y_n - f(x_n)}{s(x_n)}, [-M, M] \right), \tag{50}$$

*with conditional law $Q_n := \mathcal{L}(z_n^\star \mid \mathcal{F}_{n-1})$. Let the ideal path-average distribution be $\overline{D}_t^\star := \frac{1}{N_t} \sum_{n=1}^{N_t} Q_n$. Let $D_\tau$ denote a* per-generation *target law for ideal residuals at generation $\tau$ (e.g., the conditional law of (50) when $x$ is drawn from the boundary set at generation $\tau$). Then for any reference generation $t$,*

$$W_1(\widehat{D}_t, D_t) \leq \underbrace{W_1(\widehat{D}_t, \overline{D}_t)}_{\text{(I) finite pool}} + \underbrace{\mathrm{StdErr}_t}_{\text{(II) standardization}} + \underbrace{\kappa_t^{\mathrm{shape}}}_{\text{(III) shape shift}} + \underbrace{\mathrm{Drift}_t}_{\text{(IV) drift}}, \tag{51}$$

*where*

$$\text{StdErr}_t := W_1(\overline{D}_t, \overline{D}_t^\star), \tag{52}$$

$$\kappa_t^{\text{shape}} := \frac{1}{N_t} \sum_{n=1}^{N_t} W_1(Q_n, D_{\tau_n}), \tag{53}$$

$$\text{Drift}_t := \frac{1}{N_t} \sum_{n=1}^{N_t} W_1(D_{\tau_n}, D_t). \tag{54}$$

*Proof of Proposition 4.* Apply the triangle inequality twice:

$$W_1(\widehat{D}_t, D_t) \leq W_1(\widehat{D}_t, \overline{D}_t) + W_1(\overline{D}_t, D_t)$$
$$\leq W_1(\widehat{D}_t, \overline{D}_t) + W_1(\overline{D}_t, \overline{D}_t^\star) + W_1(\overline{D}_t^\star, D_t).$$

This yields term (I) and the definition of $\text{StdErr}_t$ in (52). To bound the remaining term, use convexity of $W_1$ in its first argument (mixture subadditivity):

$$W_1(\overline{D}_t^\star, D_t) = W_1\left( \frac{1}{N_t} \sum_{n=1}^{N_t} Q_n, D_t \right) \leq \frac{1}{N_t} \sum_{n=1}^{N_t} W_1(Q_n, D_t).$$

For each $n$, applying the triangle inequality $W_1(Q_n, D_t) \leq W_1(Q_n, D_{\tau_n}) + W_1(D_{\tau_n}, D_t)$ and average gives (53)–(54). □

To make $\text{StdErr}_t$ explicit, suppose the stored residual uses estimated center/scale, $\widehat{z}_n = \text{clip}((y_n - \widehat{f}(x_n))/\widehat{s}(x_n), [-M, M])$, while the ideal residual is (50). Since $\text{clip}(\cdot, [-M, M])$ is 1-Lipschitz and $W_1(P, Q) \leq \mathbb{E}|X - Y|$ for any coupling $(X, Y)$ of $P, Q$, one convenient bound is

$$\text{StdErr}_t \leq \frac{1}{N_t} \sum_{n=1}^{N_t} \mathbb{E}[|\alpha_n| + |\beta_n| \cdot |\varepsilon_n| \mid \mathcal{F}_{n-1}], \tag{55}$$

where $y_n = f(x_n) + s(x_n)\varepsilon_n$, $\alpha_n = \frac{f(x_n) - \widehat{f}(x_n)}{\widehat{s}(x_n)}$ and $\beta_n = \frac{s(x_n) - \widehat{s}(x_n)}{\widehat{s}(x_n)}$. Under winsorization $|\varepsilon_n| \leq M$, the second term is bounded by $M |s - \widehat{s}|/\widehat{s}$.

### C.5. Assumptions and Diagnostics

We keep the assumptions explicit and map each to a simple diagnostic, so the theory is falsifiable. See table 3 for details.

*Table 2.* Diagnostics toolbox: mapping modeling assumptions to loggable statistics.

| Assumption / term | Diagnostic (loggable) | Failure mode indicator |
| --- | --- | --- |
| Conditional independence (or weak cross-candidate correlation) | Cross-candidate correlation of standardized residuals within a generation | Strong common random numbers / shared shocks |
| Heavy tails handled by winsorization ($|\widehat{z}| \leq M$) | Clipping saturation rate $\Pr(|\widehat{z}| = M)$ | If large, analysis applies only to winsorized noise |
| Finite pool size error | Pool size $N_t$ (and bootstrap samples $B$) | Small $N_t$ yields high Wasserstein-1 ($W_1$) estimation error |
| Shape / covariate shift | Bucket by $|\widehat{m}|$ and compare buckets via KS / $W_1$ | Strong state-dependent noise shape not captured by standardization |
| Drift / nonstationarity | Sliding-window $W_1$ between recent pool slices | Rapid drift breaks pooling across time |
| Standardization error | Split-median center stability; held-out scale CV | Poor center/scale estimates induce systematic bias |
| Anti-concentration (few near ties) | Mass of pairwise gaps near 0 | Near-ties make rank-based updates unstable |

## D. Decision-Theoretic Analysis for Probe-and-Switch

In practice, before optimization (or at a designated checkpoint), we sample a probe population from the current search distribution, evaluate each probe point twice with independent oracle noise, and compute a rank-disagreement statistic:

$$P := \frac{1}{\lambda^2} \sum_{i=1}^{\lambda} \left| r_i^{(a)} - r_i^{(b)} \right| \in [0, 1/2]. \tag{56}$$

Here, $r_i^{(a)}$ and $r_i^{(b)}$ are the ranks of the same candidate under the two independent evaluation vectors. The probe consumes $2\lambda$ evaluations, reducing the remaining optimization budget to $B - 2\lambda$. Then, given a threshold $\tau \in [0, 1/2]$, we run RB-PEM if $P \geq \tau$ and otherwise run standard CMA-ES (with one evaluation per candidate and no bootstrap). This policy is explicitly budgeted: it pays a fixed probe cost up front, and it pays the RB-PEM overhead $K_t$ only in regimes where the probe indicates that intra-generation ranking uncertainty is substantial.

After paying a probe cost of $2\lambda$ evaluations, we must choose one of two actions $a \in \{0, 1\}$ for the remaining budget: $a = 0$ runs CMA-ES, and $a = 1$ runs RB-PEM. Let $L_a$ denote the random loss (e.g., final noise-free best value) obtained by action $a$ under the remaining budget, and let $P \in [0, 1/2]$ be the probe statistic (Eq. (56)). A decision rule is any measurable map $\pi : [0, 1/2] \to \{0, 1\}$, and its risk is $\mathcal{R}(\pi) = \mathbb{E}[L_{\pi(P)}]$.

**Proposition 5.** *Among all decision rules that depend only on $P$, the Bayes-optimal rule is*

$$\pi^\star(p) = \mathbf{1}\{\Delta(p) \geq 0\}. \tag{57}$$

*If $\Delta(\cdot)$ is continuous and has a single crossing, i.e., there exists $p^\star$ such that $\Delta(p) \leq 0$ for $p \leq p^\star$ and $\Delta(p) \geq 0$ for $p \geq p^\star$, then $\pi^\star$ is a threshold rule $\pi^\star(p) = \mathbf{1}\{p \geq p^\star\}$.*

*Proof.* By iterated expectation, $\mathcal{R}(\pi) = \mathbb{E}[\mathbb{E}[L_{\pi(P)} \mid P]]$. For each realized $P = p$, the inner conditional expectation is minimized by choosing the action with smaller conditional expected loss. This yields $\pi^\star(p) = \mathbf{1}\{\Delta(p) \geq 0\}$. If $\Delta(\cdot)$ single-crosses, the set $\{p : \Delta(p) \geq 0\}$ is an interval $[p^\star, 1/2]$, hence $\pi^\star$ is a threshold. $\square$

A sufficient structural condition for a monotone/threshold rule is that (i) the relative benefit of RB-PEM increases with an underlying misranking-severity parameter, and (ii) the probe statistic $P$ is stochastically increasing in that parameter (e.g., a monotone likelihood ratio signal). In our setting, $P$ is a rank-disagreement proxy that is constant-factor equivalent to Kendall discordance (Diaconis & Graham, 1977), and our mechanism analysis links higher misranking to larger update dispersion and curvature loss. We empirically validate approximate single-crossing behavior in our core COCO regime.

## E. Additional Theory Details

*Proof of Lemma 2.* For any $\mathcal{G}$-measurable $a$,

$$\mathbb{E}\left[\|\Delta m - a\|^2 \mid \mathcal{G}\right] = \mathbb{E}\left[\|\Delta m - \mathbb{E}[\Delta m \mid \mathcal{G}]\|^2 \mid \mathcal{G}\right] + \|a - \mathbb{E}[\Delta m \mid \mathcal{G}]\|^2,$$

by $\mathbb{E}[(\Delta m - \mathbb{E}[\Delta m \mid \mathcal{G}])^\top (a - \mathbb{E}[\Delta m \mid \mathcal{G}]) \mid \mathcal{G}] = 0$. $\square$

Lemma 2 is the same "conditioning improves an estimator" principle behind the classical Rao–Blackwell theorem; see, e.g., Lehmann & Casella (1998); Casella & Berger (2002). Here the $\sigma$-field $\mathcal{G}$ is generated by the candidate set (and hence contains all information unrelated to evaluation noise), so $\Delta m_{\text{PEM}}$ is the variance-reduced update that averages only over ranking uncertainty.

*Proof of Theorem 1.* Since $f$ is $\alpha$-strongly convex on the convex set $\mathcal{C}$, for any $u, v \in \mathcal{C}$,

$$f(u) \geq f(v) + \langle \nabla f(v), u - v \rangle + \frac{\alpha}{2} \|u - v\|^2. \tag{58}$$

Apply (58) with $u = X$ and $v = \bar{X}$, which is valid by Assumption 1(ii), and take conditional expectation:

$$\mathbb{E}[f(X) \mid x_{1:\lambda}] \geq f(\bar{X}) + \langle \nabla f(\bar{X}), \mathbb{E}[X - \bar{X} \mid x_{1:\lambda}] \rangle + \frac{\alpha}{2} \mathbb{E}[\|X - \bar{X}\|^2 \mid x_{1:\lambda}].$$

The middle term vanishes because $\mathbb{E}[X - \bar{X} \mid x_{1:\lambda}] = 0$ by the definition of $\bar{X}$. $\square$

| ID | Figure/Table | What it demonstrates (and why it matters) |
|---|---|---|
| F1 | Fig. 7 | A controlled mechanism chain: more misranking $\Rightarrow$ larger update dispersion $\Rightarrow$ larger curvature-induced loss, matching Theorem 1. |
| F2 | Fig. 8; Tab. 4 | RB-PEM gains are robust to estimator knobs and to the bootstrap-internal weight map, and persist even when boundary reevaluation overhead is disabled ($K_{\max} = 0$), ruling out "hidden resampling" or a special weight parametrization as the main driver. |
| F3 | Fig. 9 | Online diagnostics separate successes from failures and reveal *which* assumptions break; failures often manifest as *depth collapse* due to excessive boundary reevaluations. |
| F4 | Fig. 10 | The rank-disagreement statistic used by the probe is tightly related to Kendall discordance and top-$\mu$ elite disagreement; empirically, the theoretical bounds are never violated. |
| F5 | Fig. 11 | A counterexample where local variance is *zero* but misranking is large; misranking-based probe succeeds while variance-based probe fails, justifying our probe design. |
| F6 | Fig. 12 | Calibration curves: $\Pr(\text{RB-PEM wins} \mid P)$ is monotone in the probe statistic $P$, supporting a threshold rule as in Prop. 5. |
| F7 | Fig. 13 | Probe budget vs. reliability: small probes already discriminate well (AUC $\approx 0.71$ at $\lambda{=}4$) and saturate near the default population size. |
| F8 | Fig. 14 | Threshold choice is not brittle: accuracy and decision regret exhibit a broad plateau; misranking probe dominates a variance proxy across thresholds. |
| F9 | Fig. 15 | Depth–fidelity robustness across budgets/dimensions and a UH-CMA-ES sweep showing evaluation-stage reevaluation is structurally disadvantaged under fixed budget. |
| F10 | Fig. 16 | External validity on a nonconvex real-data task; warmstarting can eliminate probe overhead when evaluations are effectively deterministic. |
| F11 | Tab. 5 | Complete COCO results on the high-misranking function class, demonstrating the main-text pattern is not cherry-picked. |
| F12 | Tabs. 6–7 | All-function breakdowns by probe-defined regime and noise family, showing where selection-stage smoothing helps and where probe-and-switch avoids unnecessary overhead. |

*Table 3.* **Appendix experiment roadmap.** Each experiment targets a distinct link in the paper's argument: (i) depth-over-fidelity mechanism (F1, F9, F11), (ii) RB-PEM validity and robustness (F2–F4), and (iii) probe-and-switch design and tuning (F5–F8, F10).

## F. Additional Experiments

This appendix provides supplementary experimental evidence supporting the paper's central thesis: under strict fixed evaluation budgets, the dominant efficiency bottleneck for noisy rank-based evolution strategies is often *update quality per oracle call*, not update quality per generation. Accordingly, methods that spend budget on *evaluation-stage* denoising (e.g., uniform resampling or uncertainty-handling reevaluations) can lose by collapsing *depth* (the number of completed generations), while *selection-stage* uncertainty integration (RB-PEM) can improve performance at near-maximal depth. We also provide targeted evidence for the design of our probe-and-switch policy.

Note that relative to the earlier experimental plan, we (i) removed the planned transfer-matrix experiment, and (ii) merged the prior "budget grid" and "UH `maxevals` sweep" into a single robustness study. To avoid gaps and to make cross-referencing unambiguous, we renumber the appendix experiments sequentially as F1–F11 below.

Table 3 summarizes the role of each appendix experiment and the key alternative explanation it addresses.

### F.1. Mechanism Validation on a Controlled Quadratic

Theorem 1 formalizes the key mechanism behind "depth over fidelity": under local curvature, conditional update dispersion incurs an unavoidable Jensen gap, so reducing dispersion can improve expected progress per oracle call. This experiment tests the full mechanism chain in the simplest setting where the curvature penalty is analytically transparent.

**Setup**. We use a strongly convex quadratic $f(x) = \frac{1}{2}\|x\|^2$ (strong convexity parameter $\alpha = 1$) and sample $\lambda = 16$ candidates in $d = 40$ dimensions (truncation $\mu = 8$). For each of 200 independently sampled candidate sets, we draw *two* independent noisy evaluation vectors (with additive Gaussian noise of standard deviation $\sigma_{\text{noise}} = 1.0$), producing two induced rankings. We measure: (i) a two-draw rank-disagreement score $M_{\text{RD}}$ (same structure as the probe statistic $P$ in Eq. (56)); (ii) *update dispersion* $\|\Delta m^{(a)} - \Delta m^{(b)}\|^2$ under the two rankings; and (iii) the *curvature loss* (Jensen gap) $\mathbb{E}[f(m + \Delta m)] - f(m + \mathbb{E}[\Delta m])$, estimated via Monte Carlo (256 draws per candidate set).

**Results and interpretation**. Figure 7 verifies both links in the chain. First, dispersion increases with misranking (Pearson $r = 0.45$; Fig. 7a), showing that ranking noise is directly converted into update randomness by truncation selection. Second, the curvature loss matches the quadratic identity $\mathbb{E}[f(m + \Delta m)] - f(m + \mathbb{E}[\Delta m]) = \frac{1}{2}\mathrm{Var}(\Delta m)$ exactly (slope 1; Fig. 7b). Together, these results give a controlled, falsifiable confirmation of the theoretical story: misranking is not merely "label noise", it induces update dispersion, and under curvature that dispersion translates into real expected objective loss.

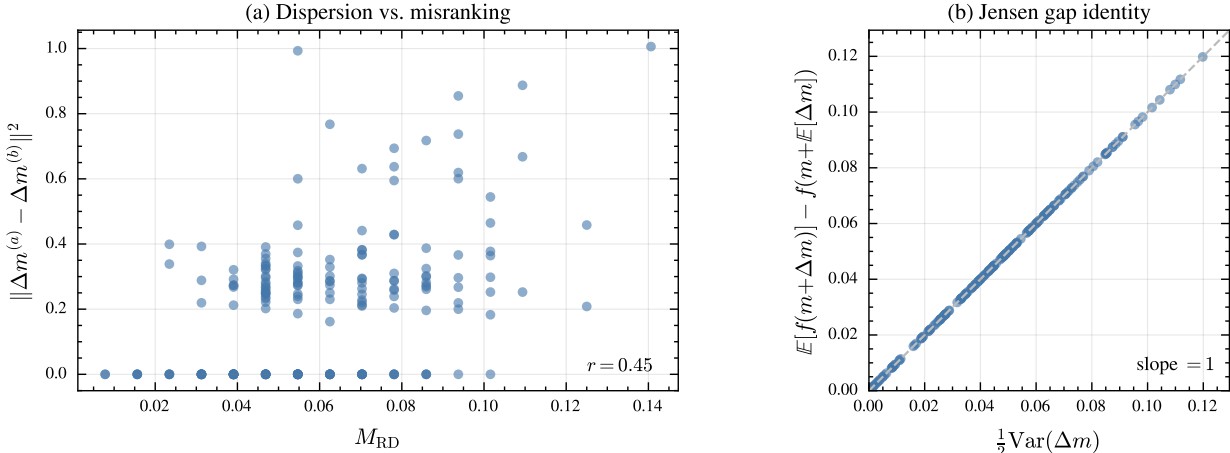

*Figure 7.* **Mechanism validation on a strongly convex quadratic.** (a) Update dispersion $\|\Delta m^{(a)} - \Delta m^{(b)}\|^2$ grows with two-draw misranking $M_{\mathrm{RD}}$ (Pearson $r = 0.45$). (b) For quadratic objectives, the Jensen gap equals $\frac{1}{2}\mathrm{Var}(\Delta m)$ exactly (slope 1), confirming that dispersion translates into expected loss under curvature. *Protocol:* $d$=40, $\lambda$=16, $\mu$=8, $\sigma_{\mathrm{noise}}$=1.0. 200 independently sampled candidate sets; 256 Monte Carlo draws per set. This is a single-step analysis (no optimization loop).

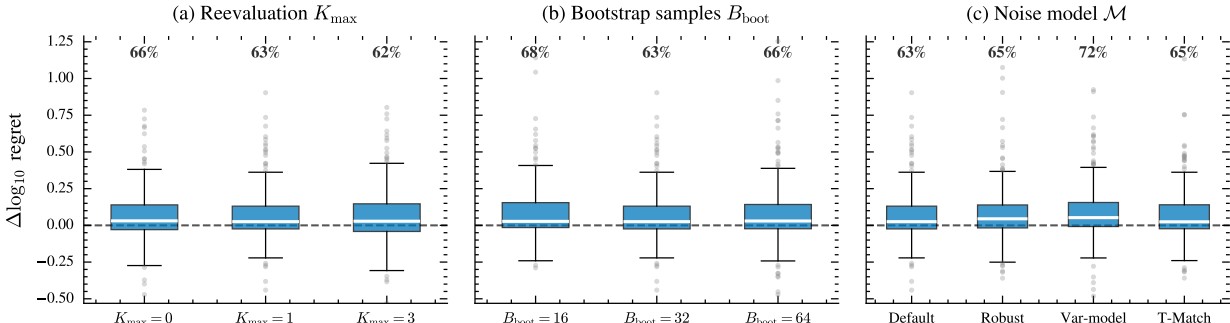

*Figure 8.* **Estimator ablations on the high-misranking COCO subset** ($d$=40, $B$=100$d$, **225 problems**). Boxplots show per-instance $\Delta \log_{10}$ regret relative to CMA-ES; the dashed line marks parity. Percentages report win rates (fraction with $\Delta > 0$). (a) Reevaluation cap $K_{\mathrm{max}}$: gains persist even at $K_{\mathrm{max}}$=0 (no boundary reevaluations). (b) Bootstrap samples $B_{\mathrm{boot}}$: stable across 16–64. (c) Noise-model choice: all variants improve; variance-based modeling yields the highest win rate (72%). *Protocol:* $\lambda$=15, $\mu$=7. Non-ablated hyperparameters held at defaults ($B_{\mathrm{boot}}$=32, $K_{\mathrm{max}}$=1). 225 problems (15 functions × 15 instances), each run once per method. Boxes show the median (center line) and interquartile range (IQR); whiskers extend to $1.5\times$ IQR; outliers plotted individually.

### F.2. RB-PEM Estimator Ablations

A natural concern is that RB-PEM's gains might be fragile (driven by a narrow hyperparameter choice), or they might simply reflect spending extra evaluations near the truncation boundary (i.e., implicit resampling). This ablation isolates these possibilities.

**Setup**. We evaluate RB-PEM variants on the COCO `bbob-noisy` high-misranking subset at $d = 40$ and budget $B = 100d$ (225 problems: 15 functions × 15 instances). For each problem instance, we report $\Delta \log_{10}$ regret $= \log_{10}(f(\hat{x}_B) - f^\star)_{\mathrm{CMA}} - \log_{10}(f(\hat{x}_B) - f^\star)_{\mathrm{variant}}$, so $\Delta > 0$ means the variant improves over CMA-ES. Figure 8 sweeps: (a) boundary reevaluation cap $K_{\mathrm{max}} \in \{0, 1, 3\}$; (b) bootstrap sample count $B_{\mathrm{boot}} \in \{16, 32, 64\}$; (c) noise-model variants used inside the bootstrap weight estimator.

**Results and interpretation**. All variants exhibit positive median improvements and clear majority win rates (62–72%;

Fig. 8). Two implications are especially important. First, the improvement persists at $K_{\max} = 0$ (66% win rate), where RB-PEM operates with essentially the same per-generation evaluation cost as vanilla CMA-ES. This strongly disfavors the hypothesis that RB-PEM wins primarily by spending extra evaluations at the boundary. Instead, it supports the intended interpretation: the main gain comes from how uncertainty is used (selection-stage smoothing of the rank weights), not from brute-force denoising. Second, performance is stable across bootstrap counts and across several plausible noise models, suggesting that the estimator is not brittle and that modest Monte Carlo effort (e.g., $B_{\text{boot}} \approx 32$) is sufficient.

### F.3. Log-Weight Ablation

The CMA-ES baseline uses its standard logarithmic recombination weights throughout. Our default RB-PEM estimator uses a smooth power-lift weight map inside the bootstrap, and Probe-and-Switch inherits this default when it switches to RB-PEM. This ablation asks whether the gains depend on that bootstrap-internal choice. We rerun matched variants that replace only the bootstrap-internal map with the standard CMA-ES log weights from (31), holding seeds, candidate populations, budgets, and all other hyperparameters fixed.

| Method (vs. CMA-ES) | $B = 20d$ | $B = 50d$ | $B = 100d$ | $B = 200d$ |
|---|---|---|---|---|
| RB-PEM (power-lift) | 65.6% | 69.5% | 68.1% | 62.2% |
| RB-PEM (log-weight) | 63.7% | 66.4% | 67.7% | 62.2% |
| Probe-and-Switch (power-lift) | 63.7% | 67.3% | 65.2% | 65.3% |
| Probe-and-Switch (log-weight) | 61.2% | 62.4% | 65.3% | 62.5% |

*Table 4.* **Log-weight ablation.** Win rates against CMA-ES on the high-misranking COCO subset across four budgets. The CMA-ES baseline uses standard logarithmic recombination weights in every row; the ablation changes only the bootstrap-internal weight map used by RB-PEM, either directly or inside Probe-and-Switch. The maximum gap between each default method and its log-weight counterpart is 3.1 percentage points, indicating that the expected-weight mechanism is not tied to the power-lift parametrization. *Protocol:* 15 high-misranking functions $\times$ 15 instances $\times$ 3 dimensions = 675 matched problems per budget; $B_{\text{boot}}$=32, $K_{\max}$=1, and shared seeds/candidate populations across paired variants.

Direct head-to-head comparisons also show near parity between the two bootstrap-internal weight maps: RB-PEM (log-weight) wins against RB-PEM (power-lift) on roughly $44\%$ of matched high-misranking problems, with median regret differences around $10^{-3}$. Median wall-clock overheads at $d = 40$, $B = 100d$ are similar as well: $1.70\times$, $1.64\times$, $1.66\times$, and $1.62\times$ CMA-ES for RB-PEM (power-lift), RB-PEM (log-weight), Probe-and-Switch (power-lift), and Probe-and-Switch (log-weight), respectively.

### F.4. Residual-Pool Diagnostic Snapshots

RB-PEM relies on reusing a pooled residual distribution; Appendix C shows that residual-pool mismatch can be decomposed into multiple falsifiable components. This experiment asks whether our diagnostics meaningfully distinguish successful runs from failures and whether they identify the relevant mismatch mode.

**Setup.** On the same 225 high-misranking COCO problems ($d = 40$, $B = 100d$), we compare RB-PEM against a representative evaluation-stage baseline (UH-CMA-ES with a conservative reevaluation budget). We label each run as *good* if RB-PEM achieves lower final regret and *bad* otherwise (193 good, 32 bad). We then compare four per-run diagnostic summaries (defined in Appendix C): drift $W_1$, shape $W_1$, scale $R^2$, and a centering-stability metric ("center rel").

**Results: diagnostics isolate specific failure modes.** Figure 9a shows that *shape* mismatch and *centering* instability are the dominant distinguishers between good and bad runs: one-sided Mann–Whitney tests detect significant separation for shape $W_1$ ($p = 0.003$) and center rel ($p = 0.001$), while drift and scale do not separate at conventional levels. This selective pattern is desirable: rather than firing a generic alarm, the diagnostics point to a concrete assumption violation ("residuals are well-standardized and shape-stable") as the likely culprit in failures.

**Results: failures often manifest as depth collapse.** Figure 9b shows per-generation traces of shape $W_1$ for a representative good and bad case. The bad case exhausts its fixed budget by generation 129, while the good case continues to generation 211 ($\Delta T = 82$ generations; 39% of depth lost). This illustrates a concrete failure pathway that is consistent with the paper's depth accounting: instability in standardization can increase boundary reevaluations (raising $K_t$), which directly consumes budget and eliminates the depth advantage RB-PEM is designed to preserve.

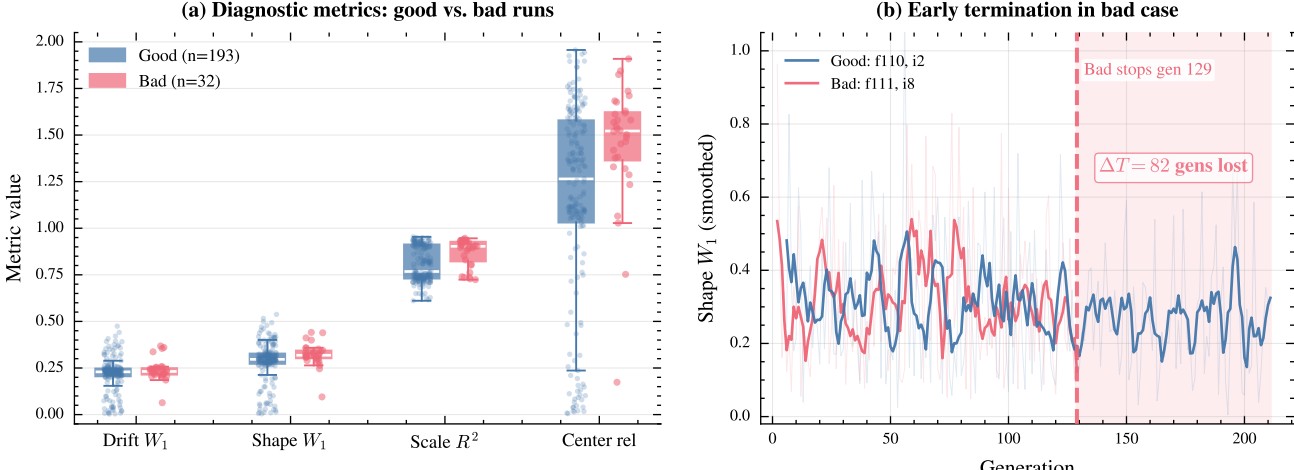

*Figure 9.* **Diagnostics make residual-pool assumptions falsifiable.** (a) Boxplots of four diagnostic summaries for *good* runs ($n = 193$, RB-PEM wins) vs. *bad* runs ($n = 32$, RB-PEM loses) on the COCO high-misranking subset ($d = 40$, $B = 100d$). Shape $W_1$ ($p = 0.003$) and centering stability ($p = 0.001$) significantly separate the two groups (one-sided Mann–Whitney); drift and scale do not, indicating that shape mismatch and standardization instability are the dominant failure modes. (b) Per-generation traces of shape $W_1$ (faint: raw; solid: smoothed) for a representative good and bad run. The bad run terminates at generation 129, losing $\Delta T = 82$ generations (39% of depth) relative to the good run (generation 211), illustrating how pool mismatch triggers depth collapse—the same mechanism the paper's budget accounting predicts. *Protocol:* $\lambda$=15, $\mu$=7, $B_{\text{boot}}$=32, $K_{\max}$=1. 225 problems (15 functions × 15 instances), each run once. Boxes show the median and IQR; whiskers extend to $1.5\times$ IQR; boxplot outlier markers suppressed. Individual data points overlaid as jittered scatter.

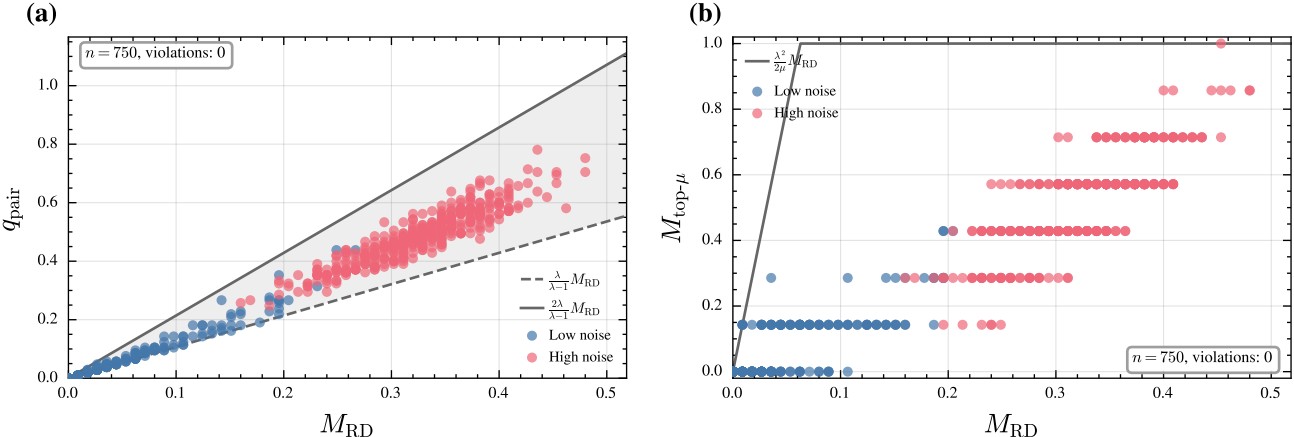

*Figure 10.* **Empirical validation of sandwich bounds for rank disagreement.** (a) Kendall discordance $q_{\text{pair}}$ vs. $M_{\text{RD}}$; (b) top-$\mu$ disagreement $M_{\text{top}\mu}$ vs. $M_{\text{RD}}$. Grey lines indicate the bounds in Eq. (59). Zero violations. *Protocol:* $d$=40, $\lambda$=15, $\mu$=7. 750 candidate sets from COCO `bbob-noisy` (30 functions × 1 instance × 25 candidate sets per function), sampled via CMA-ES evolution.

### F.5. Interpreting the Rank-Disagreement Probe via Sandwich Bounds

Probe-and-switch (Appendix D) uses a rank-disagreement statistic $P = \frac{1}{\lambda^2} \sum_i |r_i^{(a)} - r_i^{(b)}|$ (Eq. (56)). This statistic is essentially a normalized Spearman footrule distance between two permutations. To connect it to more familiar notions of misranking, we relate it to: (i) Kendall discordance (fraction of discordant pairs) and (ii) *elite disagreement* (how often the top-$\mu$ set changes), which directly governs truncation selection.

**Theoretical relations**. Let $q_{\text{pair}} \in [0, 1]$ be the fraction of discordant pairs between the two induced rankings (Kendall), and let $M_{\text{top}\mu} \in [0, 1]$ be the fraction of indices whose membership in the top-$\mu$ set differs. Standard inequalities between footrule and Kendall distances imply constant-factor bounds of the form

$$\frac{\lambda}{\lambda - 1} M_{\text{RD}} \ \leq \ q_{\text{pair}} \ \leq \ \frac{2\lambda}{\lambda - 1} M_{\text{RD}}, \qquad M_{\text{top}\mu} \ \leq \ \frac{\lambda^2}{2\mu} M_{\text{RD}}, \tag{59}$$

where $M_{\mathrm{RD}}$ is the two-draw rank-disagreement score (numerically the same object as $P$ up to notation).

**Empirical validation**. We sample 750 candidate sets from COCO `bbob-noisy` at $d = 40$ with the default $\lambda = 15$ and $\mu = 7$. Figure 10 shows $(q_{\mathrm{pair}}, M_{\mathrm{top}\mu})$ versus $M_{\mathrm{RD}}$ together with the bounds in Eq. (59). There are zero violations out of 750 points. This supports two practical interpretations: (i) $M_{\mathrm{RD}}$ is a scale-consistent proxy for overall rank instability (Kendall discordance), and (ii) large $M_{\mathrm{RD}}$ implies nontrivial instability of the top-$\mu$ set, which is precisely the part of the ranking that affects CMA-ES updates most strongly.

### F.6. Variance Does Not Equal Misranking (A Counterexample)

A tempting simplification would be to trigger switching based on a *variance probe* (estimate the noise variance at a point and switch when it is large). This experiment constructs a clean counterexample showing that variance at a single location can be arbitrarily misleading for rank-based ES, while rank disagreement remains reliable.

**Radial (state-dependent) noise model**. We consider heteroscedastic noise with scale increasing with distance to an initial reference point $x_0$: $\sigma_{\mathrm{eff}}(x) = \sigma\|x - x_0\|_{\mathrm{RMS}}$. Under this model, evaluations *at $x_0$* are effectively noiseless, but ES candidates sampled away from $x_0$ experience substantial noise—exactly where misranking matters.

**Probe decoupling**. Figure 11a shows the key pathology: the variance probe stays at machine precision across all 54 problem instances (it never triggers), while the rank-disagreement probe $M_{\mathrm{RD}}$ spans a wide range and triggers on 49/54 problems (at $\tau = 0.12$). Thus, variance can be *decoupled* from the misranking regime experienced by the candidate population.

**Algorithmic consequence**. This decoupling changes decisions. Figure 11b compares probe-and-switch using misranking detection (MR) versus a variance-based variant (Var). Misranking-based switching wins on 39/54 problems ($p = 0.0007$, sign test). This provides direct support for our design choice in Appendix D: a probe should measure ranking instability *in the current search distribution*, not variance at an arbitrary point.

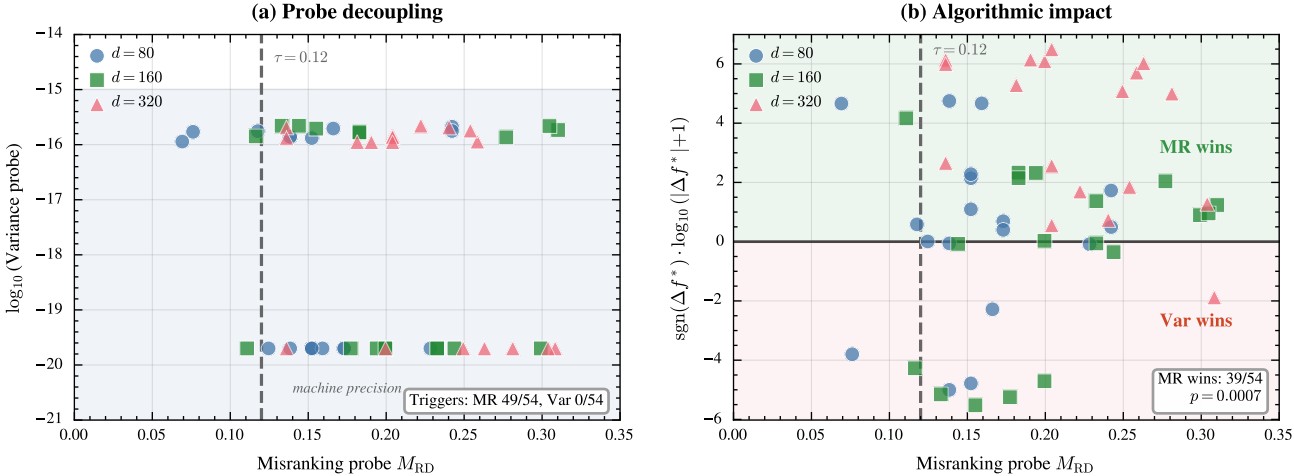

*Figure 11.* **Variance $\neq$ misranking under radial noise.** (a) The variance probe is near machine precision while the misranking probe varies widely; triggers: MR 49/54, Var 0/54. (b) Switching decisions: MR-based probe-and-switch outperforms variance-based (MR wins 39/54; $p = 0.0007$). *Protocol:* Radial noise $\sigma_{\mathrm{eff}}(x) = 0.5\|x - x_0\|$, $d \in \{80, 160, 320\}$, $B = 200d$, $B_{\mathrm{boot}}=32$, $K_{\max}=1$. $\lambda$ follows the CMA-ES default at each $d$ (17, 19, 21). 54 problems (6 functions $\times$ 3 instances $\times$ 3 dimensions), each run once per method.

### F.7. Probe Calibration Curves

Proposition 5 shows that if the conditional advantage $\Delta(p)$ single-crosses, then a *threshold* rule in the probe statistic $P$ is Bayes-optimal among rules depending only on $P$. This experiment empirically checks whether $P$ behaves as a calibrated predictor of when RB-PEM wins.

**Setup**. We compute the probe statistic $P$ (Eq. (56)) and bin problems into quantiles of $P$. Within each bin we estimate the empirical win rate of RB-PEM over CMA-ES, with 95% Wilson score intervals. To test generalization and avoid within-suite leakage, we evaluate on a held-out test split of instances (instances 6–15), using thresholds calibrated on instances 1–5. We report two representative budgets to illustrate budget dependence.

**Results and interpretation**. Figure 12 shows a clear monotone trend: at low $P$ (stable ranks), RB-PEM wins with probability well below 0.5 (CMA-ES is preferred); at high $P$, RB-PEM wins with probability 0.7–0.8. The calibrated thresholds (vertical dashed lines) fall near the empirical 0.5 crossing, aligning with the decision-theoretic interpretation: the probe is measuring a quantity that is predictive of the algorithm crossover. The threshold shifts with budget in the expected direction: when the remaining budget is larger, the relative cost of RB-PEM's bounded overhead is smaller, so switching becomes beneficial at lower misranking levels.

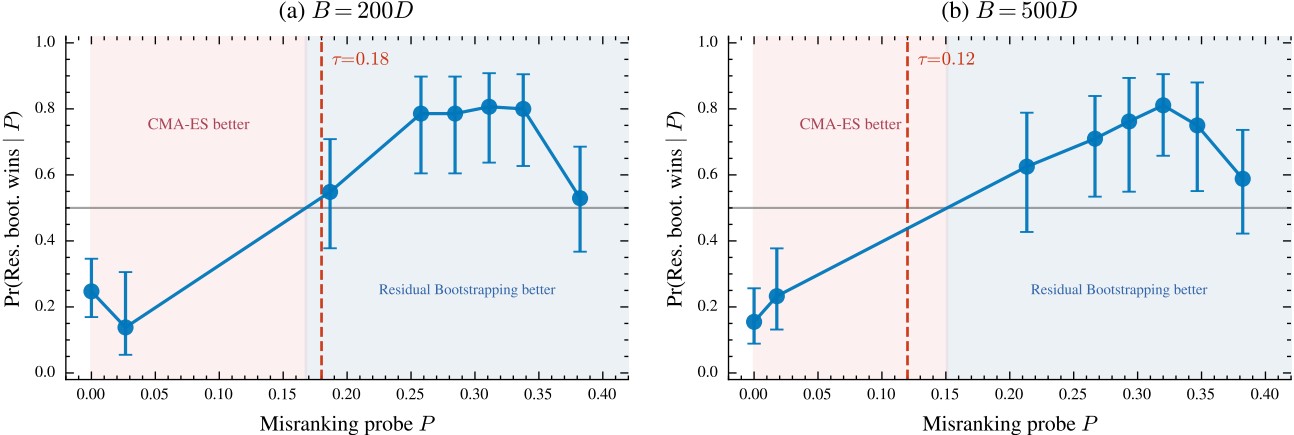

*Figure 12.* **Probe calibration curves.** Empirical $\Pr(\text{RB-PEM wins} \mid P)$ versus probe statistic $P$ (quantile bins; 95% Wilson intervals). Shaded regions indicate which algorithm is preferred; the vertical dashed line is the calibrated threshold. The win probability increases monotonically with $P$, supporting a threshold decision rule. *Protocol:* $d=40$, $\lambda=15$, $\mu=7$, $B_{\text{boot}}=32$, $K_{\text{max}}=1$. Two budget levels: $B=200d$ and $B=500d$. 450 problems per budget (30 functions $\times$ 15 instances), each run once. Calibration uses instances 1–5 (train) and 6–15 (test).

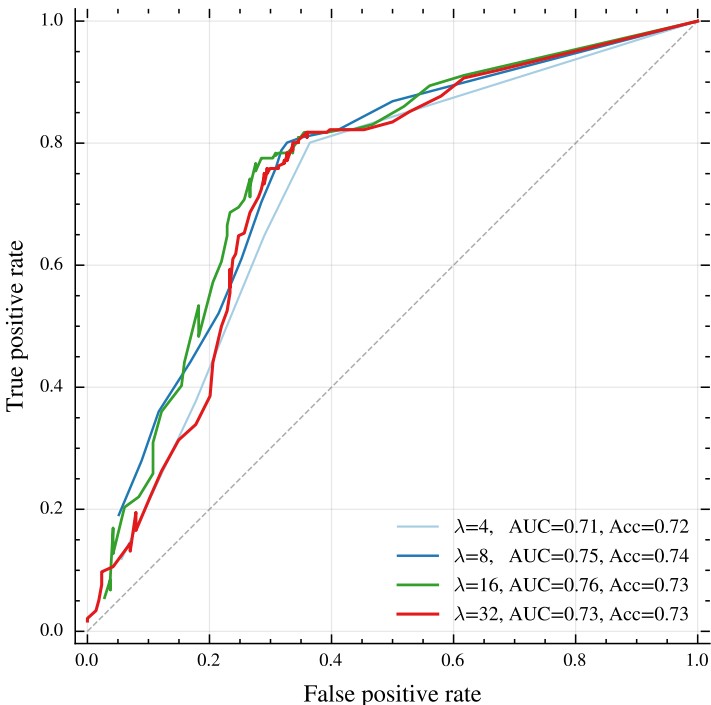

*Figure 13.* **Probe budget vs. ROC.** ROC curves for different probe population sizes $\lambda$ on COCO `bbob-noisy` ($d=40$, $B=200d$). Legend: AUC and accuracy at the displayed operating point. Reliability improves up to $\lambda \approx 16$ and then saturates. *Protocol:* $B_{\text{boot}}=32$, $K_{\text{max}}=1$. 30 functions $\times$ 15 instances = 450 problems per $\lambda$ value, each run once. $\lambda \in \{4, 8, 16, 32\}$; $\mu=\lfloor\lambda/2\rfloor$. Operating point: $\tau=0.12$.

## F.8. Probe Reliability versus Probe Budget

The probe costs $2\lambda$ evaluations, so it must be informative at modest budgets. This experiment quantifies the classification quality of the probe as we vary its cost by changing $\lambda$.

**Setup**. We sweep probe population sizes $\lambda \in \{4, 8, 16, 32\}$. For each setting we compute $P$ and evaluate its ability to classify problems where RB-PEM beats CMA-ES (ground-truth labels from full runs) on 450 COCO problems at $d = 40$, $B = 200d$. We report ROC curves; the legend lists AUC and accuracy at a representative threshold.

**Results and interpretation**. Figure 13 shows that even the cheapest probe ($\lambda = 4$, cost 8 evaluations) is strongly informative (AUC 0.71, accuracy 0.72), and performance improves up to $\lambda \approx 16$ (AUC 0.76). Increasing probe cost further does not help (AUC drops to 0.73 at $\lambda = 32$), indicating diminishing returns and a clear cost-effectiveness plateau. This justifies using a modest probe budget comparable to the default ES population size.

## F.9. Threshold Sensitivity Analysis

Probe-and-switch introduces a single scalar hyperparameter: the switching threshold $\tau$. A practical method should not require fragile tuning. This experiment evaluates (i) how accuracy depends on $\tau$ and (ii) how much regret is incurred by choosing a suboptimal threshold.

**Setup**. We sweep $\tau \in [0, 0.30]$ and evaluate classification accuracy of the misranking probe (MR) and a variance proxy (Var) at two budgets. We also compute *decision regret*: the mean $\log_{10}$ performance gap to an oracle that always chooses the better of CMA-ES and RB-PEM for each problem instance. We report both the train split (instances 1–5) and the held-out test split (instances 6–15).

**Results and interpretation**. Figure 14a shows a broad accuracy plateau: for MR, accuracy varies by only $\approx 2$ percentage points over $\tau \in [0.08, 0.18]$. Across all thresholds, MR outperforms Var by roughly 5–6 percentage points, reinforcing the necessity of rank-based probing (cf. F5). Figure 14b shows that decision regret is highly asymmetric: very small $\tau$ causes *over-switching* (unnecessary RB-PEM overhead on low-misranking problems), while very large $\tau$ causes *under-switching* (missing beneficial RB-PEM deployments). Importantly, the regret curve is also flat across the plateau, so near-optimal decisions do not require precise threshold tuning; $\tau = 0.12$ lies safely in the robust region.

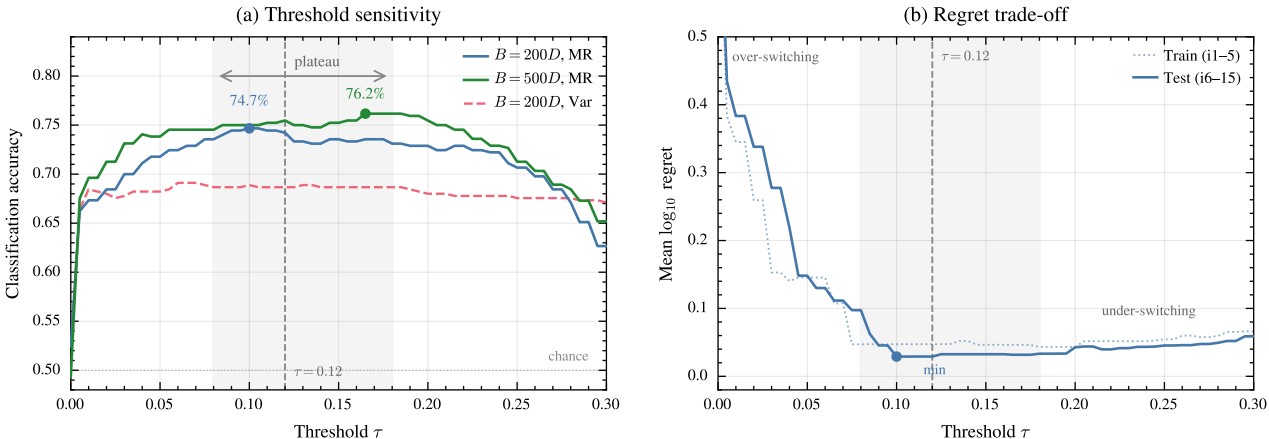

*Figure 14.* **Threshold sensitivity.** (a) Classification accuracy vs. threshold $\tau$: MR exhibits a broad plateau and outperforms Var. (b) Decision regret vs. $\tau$: over-switching dominates at small $\tau$, under-switching at large $\tau$; the minimum lies within the accuracy plateau, showing robust tuning. *Protocol:* $d$=40, $\lambda$=15, $\mu$=7, $B_{\text{boot}}$=32, $K_{\text{max}}$=1. Two budget levels: $B$=200$d$ and $B$=500$d$. 450 problems per budget (30 functions $\times$ 15 instances), each run once. Accuracy is the proportion of problems correctly routed; decision regret is the mean $\log_{10}$ performance gap to an oracle selector.

## F.10. Depth–Fidelity Robustness and UH-CMA-ES Sensitivity

The main paper argues that under fixed budget, evaluation-stage denoising is structurally disadvantaged because it reduces depth. This section provides two complementary robustness checks: (a) whether RB-PEM's advantage persists across budgets and dimensions, and (b) whether UH-CMA-ES can be "rescued" by tuning its `maxevals` parameter.

**a: Budget and dimension robustness**. Figure 15a reports win rates of RB-PEM against three baselines across budgets $B \in \{50d, 100d, 200d\}$ and dimensions $d \in \{20, 40\}$ (225 high-misranking COCO problems per setting). RB-PEM wins decisively against UH-CMA-ES (roughly 79–94%) and against Resample($k$=10) (roughly 78–83%) in every regime, and maintains majority wins even against vanilla CMA-ES. The advantage is larger at $d = 40$ than at $d = 20$, consistent with misranking becoming more damaging in higher dimensions and with the paper's empirical observation that depth becomes increasingly predictive of performance as misranking increases.

**b: UH-CMA-ES `maxevals` sweep**. Figure 15b sweeps `maxevals`$\in \{1, 10, 30\}$. Across both budgets, UH-CMA-ES remains far below the 50% parity line versus CMA-ES and versus probe-and-switch. Moreover, increasing `maxevals` monotonically *reduces* UH-CMA-ES win rate: allocating more reevaluations per generation worsens fixed-budget performance. This supports the interpretation that UH-CMA-ES's underperformance is not a tuning artifact but a depth-loss effect.

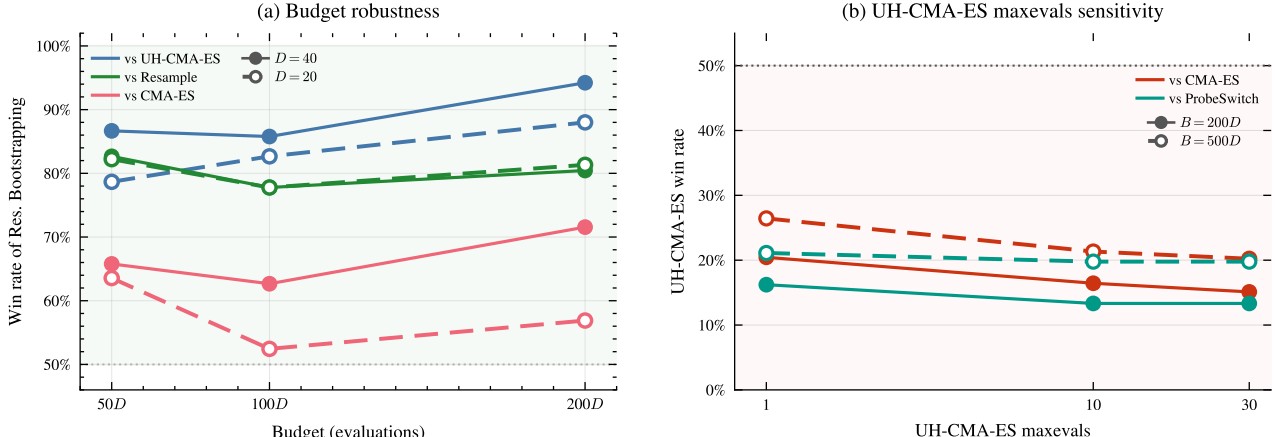

*Figure 15.* **Depth–fidelity robustness and baseline sensitivity.** (a) Win rate of RB-PEM across budgets and dimensions against UH-CMA-ES, Resample($k$=10), and CMA-ES. (b) UH-CMA-ES win rate vs. CMA-ES and vs. Probe-and-Switch ($\tau$=0.12) as a function of `maxevals`; all values are far below parity, and larger `maxevals` further degrades performance. *Protocol:* $B_{\mathrm{boot}}$=32, $K_{\max}$=1. Panel (a): $d \in \{20, 40\}$, $B \in \{50d, 100d, 200d\}$, 15 high-misranking functions × 15 instances = 225 problems per (budget, dimension) pair, each run once. Panel (b): $d$=40, $B \in \{200d, 500d\}$, 30 functions × 15 instances = 450 problems per budget, each run once; `maxevals`$\in \{1, 10, 30\}$.

### F.11. External Validity on a Nonconvex Real-Data Task

The COCO suite is synthetic and largely separable from end-to-end ML pipelines. This experiment tests whether the probe-and-switch principle generalizes to a small nonconvex real-data task where noise is induced by stochastic mini-batching.

**Setup**. We train a single-hidden-layer MLP (4 hidden units) on the `digits0` binary classification task (digit 0 vs. non-0), with $n = 256$ samples and $d = 265$ parameters. We control ranking noise via mini-batch size $B_{\mathrm{batch}} \in \{4, 16, 256\}$ (smaller batches $\Rightarrow$ noisier objective values $\Rightarrow$ more misranking). We compare CMA-ES, probe-and-switch, and a *warmstart* variant that reuses the probe evaluations as part of the first generation, thereby removing the probe's opportunity cost when the probe indicates "no switch." All methods use the same total budget $B = 40d = 10{,}600$ and 50 random seeds.

**Results and interpretation**. Figure 16 shows three regimes. *Moderate noise* ($B_{\mathrm{batch}} = 16$, $M_{\mathrm{RD}} \in [0.13, 0.34]$): probe-and-switch improves over CMA-ES (66% win rate) and warmstart is stronger still (76% win rate), consistent with the paper's main claim that selection-stage uncertainty integration is most beneficial in high-misranking regimes. *Extreme noise* ($B_{\mathrm{batch}} = 4$): both variants are near parity (54% win rate), suggesting a saturation regime where all methods are heavily noise-limited. *Deterministic* ($B_{\mathrm{batch}} = 256$, $M_{\mathrm{RD}} = 0$): warmstart matches CMA-ES almost exactly (48/50 ties, i.e., 96%), demonstrating that the probe can correctly detect "no misranking" and that warmstarting can remove probe cost when switching is unnecessary. Overall, this nonconvex experiment supports the external validity of probe-and-switch and highlights warmstarting as a simple practical refinement in low-noise settings.

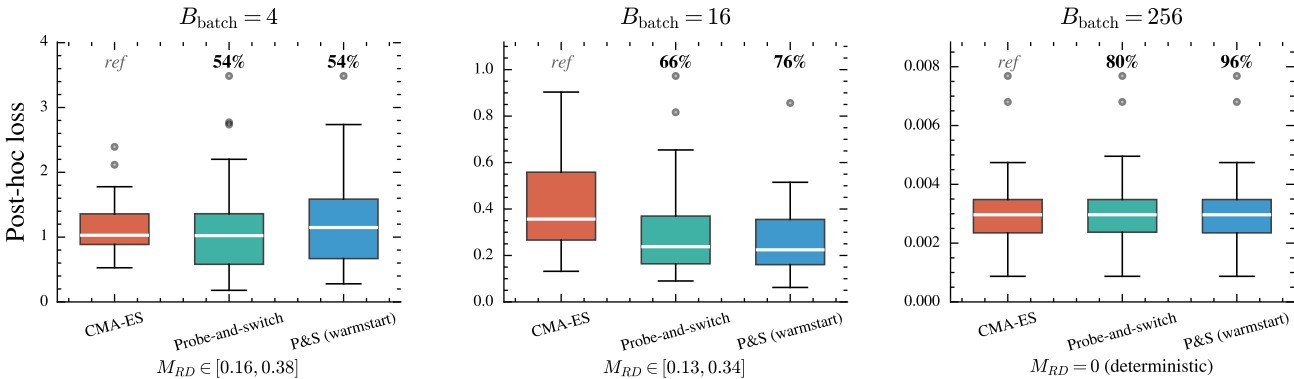

*Figure 16.* **MLP training on `digits0` (nonconvex).** Final post-hoc loss (lower is better) across 50 seeds at fixed budget $B = 40d$. Numbers above boxes indicate the fraction of seeds on which the method improves upon (or ties with) CMA-ES. Left: $B_{\text{batch}} = 4$ (extreme noise); Center: $B_{\text{batch}} = 16$ (moderate noise); Right: $B_{\text{batch}} = 256$ (deterministic). *Protocol:* $d=265$, $\lambda=20$ (CMA-ES default), $\mu=10$, $B_{\text{boot}}=32$, $K_{\max}=1$. 50 independent seeds per (method, batch-size) pair. Boxes show the median (center line) and IQR; whiskers extend to $1.5\times$ IQR; outliers plotted individually.

## F.12. Complete Results on High-Misranking COCO Functions

The main text visualizes convergence on a small set of representative functions. Table 5 provides the complete per-function summary on the full high-misranking function class used throughout the paper, to verify that the reported gains are not an artifact of selective function choice.

**Setup**. We report median final $\log_{10}$ regret at budget $B = 100d$ for $d = 40$ across 15 instances per function. We compare CMA-ES, UH-CMA-ES, RB-PEM, and probe-and-switch (bold indicates the best median per function).

**Results and interpretation**. Two patterns stand out. First, RB-PEM and probe-and-switch *dominate* UH-CMA-ES across the entire class (no exceptions), reinforcing the depth-fidelity argument: evaluation-stage reevaluation is consistently uncompetitive under strict budgets. Second, selection-stage methods usually outperform vanilla CMA-ES, with the largest gains on the functions where the noisy ranking is most unstable (e.g., f110, f111, f116). On the few functions where CMA-ES is marginally better, the gaps are small, consistent with a low-misranking regime where smoothing and any additional overhead bring limited benefit. This table therefore complements the main-text figures by showing that the "selection-stage wins" pattern is broad and not cherry-picked.

*Table 5.* **Median $\log_{10}$ regret at $B = 100d$ across all 15 high-misranking COCO functions** ($d=40$, 15 instances each). **Bold**: lowest per function. *Protocol:* $\lambda=15$, $\mu=7$, $B_{\text{boot}}=32$, $K_{\max}=1$. Probe-and-Switch uses $\tau=0.12$. Each instance is run once per method; the reported statistic is the median across 15 instances.

| Function | *Baselines* | | *Selection-stage* | |
| | CMA-ES | UH-CMA-ES | RB-PEM | Probe-and-switch |
| --- | --- | --- | --- | --- |
| f108 | 2.39 | 2.46 | 2.37 | **2.36** |
| f110 | 4.87 | 5.35 | **4.47** | 4.53 |
| f111 | 5.22 | 5.48 | 5.11 | **5.09** |
| f113 | 2.87 | 3.05 | 2.77 | **2.75** |
| f114 | **3.03** | 3.16 | 3.13 | 3.06 |
| f116 | 4.65 | 4.99 | **4.48** | 4.54 |
| f117 | 4.93 | 4.98 | 4.90 | **4.89** |
| f119 | 1.58 | 1.74 | 1.53 | **1.50** |
| f120 | 1.73 | 1.84 | 1.69 | **1.68** |
| f122 | 1.04 | 1.14 | **1.03** | 1.04 |
| f123 | **1.13** | 1.19 | 1.15 | 1.15 |
| f125 | 0.26 | 0.36 | 0.12 | **0.10** |
| f126 | 0.32 | 0.39 | 0.22 | **0.21** |
| f128 | 1.91 | 1.92 | 1.91 | **1.90** |
| f129 | 1.91 | 1.91 | 1.91 | **1.91** |

## F.13. All-Function Breakdowns by Probe Regime and Noise Family

Table 1 and Fig. 4 report all-function evidence; here we unpack where those gains arise. The probe statistic separates `bbob-noisy` into a selected high-misranking subset and a more stable complement. Table 6 quantifies this split at $B = 100d$. Always-on RB-PEM is strongest on the selected high-misranking functions but can be harmful on the complement, where smoothing and reevaluation overhead are often unnecessary. Probe-and-Switch preserves most of the high-misranking gain while reverting toward CMA-ES on the complement.

| Method (vs. CMA-ES) | Selected high-misranking 15f | Complement 15f |
|---|---|---|
| RB-PEM | 68.1% (460/675) | 35.3% (238/675) |
| RB-PEM (log-weight) | 67.7% (457/675) | 37.3% (252/675) |
| Probe-and-Switch | 65.2% (440/675) | 47.4% (320/675) |
| Probe-and-Switch (log-weight) | 65.3% (441/675) | 46.1% (311/675) |

*Table 6.* **High-misranking versus complement breakdown.** Win rates against CMA-ES at $B = 100d$ after splitting all 30 COCO `bbob-noisy` functions by the probe-based partition of Fig. 4. RB-PEM variants win decisively on the selected high-misranking subset but lose on the complement, while Probe-and-Switch stays near parity there by reverting to CMA-ES when rankings are stable. *Protocol:* $d \in \{10, 20, 40\}$, 15 instances per function, one run per method, 675 matched problems per column. The selected high-misranking column is the 15-function subset used throughout the paper; the complement contains the 14 low-misranking functions plus f107, whose large probe value is structural rather than noise-induced.

Table 7 gives an orthogonal view by COCO noise family. The moderate-Gaussian group is mostly stable-ranking, so always-on RB-PEM often pays overhead without enough ranking uncertainty to offset it. By contrast, the severe Gaussian, Cauchy, and multimodal groups contain regimes where selection-stage smoothing is more useful. The Cauchy row should be read as an empirical stress test: raw Cauchy noise violates the finite-variance assumption, while the implementation clips standardized residuals to $[-M, M]$ and therefore bootstraps from a winsorized working distribution.

| Noise family | Functions | Win/Loss | Win rate |
|---|---|---|---|
| Moderate Gaussian | f101–f109 | 40/95 | 29.6% |
| Severe Gaussian | f110–f118 | 87/48 | 64.4% |
| Severe Cauchy | f119–f124 | 55/35 | 61.1% |
| Severe + multimodal | f125–f130 | 61/29 | 67.8% |

*Table 7.* **Noise-family breakdown for RB-PEM (log-weight).** Win/loss counts and win rates against CMA-ES at $B = 100d$, grouped by the COCO `bbob-noisy` noise families. Gains concentrate in the severe Gaussian, severe Cauchy, and multimodal families, whereas the mostly stable moderate-Gaussian family is less favorable to always-on smoothing. *Protocol:* $d = 40$, 15 instances per function, one run per method. The Cauchy row is an empirical robustness check for the implemented winsorized residual pool, which clips standardized residuals to $[-M, M]$; it is not a claim that the raw Cauchy noise satisfies the finite-variance theory.

**Practical takeaway**. Across the appendix, the empirical message is consistent with the theoretical one: when ranking noise is substantial, it is more sample-efficient (under a strict budget) to integrate uncertainty into selection weights (RB-PEM) than to spend evaluations trying to eliminate it, and a low-cost rank-disagreement probe provides a robust mechanism for avoiding overhead in low-misranking regimes.

