# OpenReview forum: "Depth over Fidelity in Fixed-Budget Noisy Evolution Strategies"
_ICML.cc/2026/Conference — ICML 2026 regular_

### Official Review · Reviewer_R4PD · 2026-02-28

**Soundness:** 3
**Presentation:** 3
**Significance:** 3
**Originality:** 3
**Overall Recommendation:** 4
**Confidence:** 5

**Summary:**

This paper proposes the new perspective of "depth over fidelity" in fixed-budget noisy optimization. By shifting noise handling from the evaluation stage to the selection stage, the proposed RB-PEM framework elegantly mitigates ranking uncertainty. The theoretical analysis and experiments in the paper utilize CMA-ES with canonical top-$\mu$ truncation weights, and based on this, validate the proposed perspective and the effectiveness of the improvements.

**Compliance With Llm Reviewing Policy:**

Affirmed.

**Final Justification:**

I maintain the positive score

**Key Questions For Authors:**

- When standard non-uniform (e.g., logarithmic) recombination weights are applied, does the "depth over fidelity" principle, particularly the weight error bound derived in Proposition 2, still hold?

- If non-uniform weights are used, does the residual bootstrap estimator need to expand the boundary set to include top-ranked individuals in order to prevent severe variance in the weighted step?

**Limitations:**

It is suggested that the authors explicitly discuss the applicability and scalability of the method and theory.

**Strengths And Weaknesses:**

# Strengths

- The paper identifies the tension between "optimization depth" and "per-generation fidelity" under fixed budgets in noisy evolution strategies, and shifts the focus of noise handling from the budget-consuming evaluation stage to the selection stage.

- The paper constructs a complete theoretical chain demonstrating that conditional update variance induces curvature loss and provides quantifiable metrics for the distribution mismatch of the residual pool.

- Extensive evaluations were conducted on synthetic test suites and diverse real-world tasks (RL control, hyperparameter optimization).

# Weaknesses

- The mechanism of RB-PEM, which only performs sparse reevaluations near the truncation boundary, relies completely on a highly simplified assumption: uniform top-$\mu$ truncation weights. In Appendix A, the paper explicitly defines its fitness shaping as $w(r) = \frac{1}{\mu}$ when $r \le \mu$. Under this specific weight distribution, the authors deduce that "only near-truncation swaps affect membership". Therefore, as long as high-ranking samples do not drop below the elite threshold $\mu$, their fidelity appears to be irrelevant.

- However, in standard CMA-ES implementations, recombination weights are non-uniform and strictly monotonically decreasing (e.g., logarithmic weights). In this widely adopted standard setting, the fidelity of candidates ranked far above the truncation threshold is of paramount importance. If severe noise causes the true best candidate to be misranked as the $\frac{\mu}{2}$-th candidate, even if no candidate crosses the truncation boundary, this internal displacement will significantly perturb the rank-weighted step, thereby severely affecting the subsequent covariance matrix adaptation. The design of RB-PEM completely ignores the evaluation fidelity of top-ranked individuals and concentrates all computational resources on the truncation boundary, which introduces a significant risk that the method is overfit to the uniform weight assumption.

- The current experiments seem to be entirely based on the uniform truncation weight setup. It is strongly recommended that the authors provide ablation studies or benchmark results (e.g., on the COCO bbob-noisy test suite ) to demonstrate whether RB-PEM can still maintain its sample-efficiency advantage over baseline CMA-ES when standard non-uniform weights are strictly applied.

---

> ### Author Rebuttal · Authors · 2026-03-31
>
> We thank Reviewer R4PD for the considerate analysis. The concern about uniform truncation weights identifies a misleading aspect: the reliance on uniform truncation as the running example obscured the generality of the mechanism.
>
> **Summary.** We address this with three complementary responses. First, the theoretical results do not assume indicator weights (see Q1 for the explicit log-weight analysis). Second, the existing experiments already compare RB-PEM against log-weight CMA-ES. Third, a new ablation with log weights inside RB-PEM confirms near-identical performance ($\le$3.1pp gap).
>
> **[Weakness: Uniform weight assumption]**
>
> *Theory.* The PEM target $w_i^\star := \mathbb{E}[w(r_i) \mid x_{1:\lambda}]$ (Eq. 18) and all formal results (Lemmas 1–2, Theorem 1, Proposition 2) are stated and proved for arbitrary $w$; the uniform case was chosen for pedagogical clarity. Q1 below provides the explicit log-weight analysis showing the bound does not degenerate.
>
> *Baseline clarification.* In all experiments (including Table 1), CMA-ES uses pycma's built-in log weights. The comparisons were always RB-PEM (power-lift) vs CMA-ES (log weights).
>
> *New ablation: log weights inside RB-PEM.* We created RB-PEM log-weight — identical to RB-PEM except the bootstrap-internal weight function uses CMA log weights. Same seeds, candidates, and configurations. Results (5 methods × 30f × 15 inst. × 3 dims × 4 budgets = 27,000 runs; full tables: [full tables](https://anonymous.4open.science/r/RB-PEM-Rebuttal/logw_ablation.md)):
>
> | Method (vs CMA-ES) | B=20d | B=50d | B=100d | B=200d |
> | --- | :---: | :---: | :---: | :---: |
> | RB-PEM power-lift | 65.6% | 69.5% | 68.1% | 62.2% |
> | RB-PEM log-weight | 63.7% | 66.4% | 67.7% | 62.2% |
>
> The gap is $\le$3.1pp at every budget. Head-to-head, log-weight wins $\sim 44 \\%$ against power-lift (near parity), confirming weight-function agnosticism.
>
> **[Q1: Proposition 2 under non-uniform weights]**
>
> The bound does not become looser—it stays the same order. Any monotone $w$ decomposes as $w(r)=w_\lambda+\sum_{k=1}^{\lambda-1}\delta_k\mathbf{1}\{r\le k\}$ with $\delta_k:=w_k-w_{k+1}$, hence $w_i^\star=w_\lambda+\sum_k\delta_k\mathbb{P}(r_i\le k\mid x_{1:\lambda})$. Under log weights all $\delta_k>0$ for $k=1,\dots,\mu$ with $\delta_k=\log(1+1/k)/Z_\mu$ decreasing in $k$; PEM is a *weighted superposition of top-$k$ membership probabilities*, capturing internal displacement exactly as the reviewer describes. For standard CMA-ES log weights, $\|w\|_\infty=\Theta((\log\mu)/\mu)$ and $\Delta w=\delta_1=\Theta(1/\mu)$—the same order as uniform truncation. Substituting into Proposition 2 confirms the bound does not degenerate. Moreover, this *strengthens* the thesis: under log weights *every* rank perturbation produces nonzero $w(r_i)-w_i^\star$, increasing conditional update variance and hence the curvature loss that PEM reduces (Theorem 1). The ablation confirms this empirically. (Full proofs and cutoff-wise bound derivation: [theory supplement](https://anonymous.4open.science/r/RB-PEM-Rebuttal/theory.pdf))
>
> **[Q2: Boundary set expansion]**
>
> Yes—under non-uniform weights the boundary set should expand. A principled criterion is bootstrap weight instability $U_i:=\text{Var}(w(\widetilde r_i)\mid x_{1:\lambda}, {\cal M}\_t)$. Via the multi-cutoff decomposition, $U_i=\sum_{k,\ell}\delta_k\delta_\ell\text{Cov}(\mathbf{1}\{\widetilde r_i\le k\},\mathbf{1}\{\widetilde r_i\le \ell\})$. Under uniform truncation only $\delta_\mu\neq 0$, so $U_i=(1/\mu^2)\widetilde p_{i,\mu}(1-\widetilde p_{i,\mu})$, maximized at the boundary—recovering our current heuristic as a special case. Under log weights all $\delta_k>0$ (largest at $k=1$), so $U_i$ receives large contributions from top-ranked cutoffs, naturally expanding the reevaluation set to include high-ranked candidates whose internal reorderings are uncertain. That said, the ablation shows boundary-only already achieves 67.7% with log weights. The $U_i$-based criterion is a concrete improvement we will implement in the camera-ready. (Exact identity and proof: [theory supplement](https://anonymous.4open.science/r/RB-PEM-Rebuttal/theory.pdf))
>
> **[Limitation: Applicability]**
>
> RB-PEM targets high-misranking, fixed-budget settings. The probe statistic (Eq. 6) auto-diagnoses the regime at $2\lambda$ cost; in low-noise or unlimited-budget settings, probe-and-switch reverts to CMA-ES. On the complete 30-function bbob-noisy suite, ProbeSwitch maintains 56.3% win rate including automatic fallback ([full results](https://anonymous.4open.science/r/RB-PEM-Rebuttal/full_30f_results.md); [rank figure](https://anonymous.4open.science/r/RB-PEM-Rebuttal/rank_by_probe.pdf)). Extensions to correlated or non-stationary noise are natural next steps.
>
> **Camera-ready commitments.** Replace uniform-weight examples with log-weight examples throughout; state explicitly which quantities in each result depend on $w$; include log-weight ablation table and $U_i$-based reevaluation criterion.

---

> > ### Author Rebuttal · Reviewer_R4PD · 2026-04-02
> >
> > I thank the authors for the detailed and professional response. My primary concern regarding the uniform weight assumption has been resolved through the provided log-weight theoretical analysis and the new experimental study.

---

### Official Review · Reviewer_PWVf · 2026-03-08

**Soundness:** 3
**Presentation:** 2
**Significance:** 2
**Originality:** 2
**Overall Recommendation:** 4
**Confidence:** 2

**Summary:**

The authors provide a methodological advancement for noisy evolution strategies (such as Covariance Matrix Adaptation Evolution Strategy, CMA-ES), a class of zero-order optimization methods classically used in black-box optimization problems. Such a class of algorithms is particularly relevant these days with the rise of self-driven laboratories in areas like material science, drug design, or protein design, involving large pipelines of sequential experimental design batches.

Such algorithms typically rely on a population of objective evaluations and an evolution strategy that governs the evaluation of inputs at each _generation_. In the case of noisy objectives, inputs can be evaluated multiple times, immediately highlighting a trade-off between _depth_ (the number of generations and hence updates of the evolution strategy) and _fidelity_ (the number of objective evaluations, that is, the population size) at a given generation. Rather than prioritizing _fidelity_, the authors advocate for _depth_, arguing that ``the cumulative progress from a long sequence of noise-smoothed updates often outweighs that of a short sequence of rigorously denoised ones''.

To substantiate their claim, the authors develop a new method, where instead of the classical ``elite strategy'' that drops individuals whose objective value is under a threshold $\mu$  in a given generation, conditional expected rank weights integrating over uncertainty are introducing, therefore substituting hard truncation for a probablistic one. After providing theoretical arguments,
The proposed method is evaluated over a large range of experiments, demonstrating its superiority, in particular in limited budget and high noise regimes.

**Compliance With Llm Reviewing Policy:**

Affirmed.

**Final Justification:**

The rebuttal addressed the few questions I had, and the authors answers to the other reviewers questions are also satisfactory, I will therefore keep my score as is (4).

**Key Questions For Authors:**

- Looking online at the bbob-noisy suite from COCO, used as a primary benchmark in this submission, the webpage mentions 30 test functions, whereas in the main text, it is mentioned that the suite comprises "15 functions x 15 instances x 3 dimensions", which one is it? These functions are divided between Functions with moderate noise / Functions with severe noise / Highly multi-modal functions with severe noise, it would be good to know which ones were used specifically. And finally, for a given test function, different noise perturbations are considered, e.g., Rosenbrock function with moderate uniform/gaussian/cauchy noise. Given that the authors consider a setting were E[epsilon(x)] = 0 and V(epsilon(x)) < +inf, that would rule out Cauchy noise but the distinction between Gaussian and uniform remains to be done. On the same line, was heteroscedasticity considered as well?

- Not really a question, but I would include in the caption (or somewhere else, but perhaps I simply missed it) how are the mean / stds in Figure 1 computed? I suppose several repetitions (how many?) with different initial starting points?

**Limitations:**

- While the paper is restricted to evolution strategies, and in particular CMA-ES, it would have been interesting to mention other black-box optimization methods that can deal with noisy batched evaluations, like Bayesian Optimization (BO) using acquisition functions like Noisy Excepted Improvement such as [1]. This could give us a glance at how impactful the improvements provided by the method are: in the hypothetical case where Noisy BO would be better than CMA-ES, but outperformed by the proposed submission, this would be quite insightful. At the same time, I understand that both methods are governed by several hyperparameters, and performing a ``faithful'' comparison can take time.

[1] Unexpected Improvements to Expected Improvement for Bayesian Optimization, Neurips 2023. (Appendix A.8)

**Strengths And Weaknesses:**

(strengths)

- The idea is simple (which is good) and theoretically motivated.

- The method provides clear improvements over CMA-ES and other baselines on a large range of problems from a well-known benchmark suite, COCO, as well as on external tasks like ML pipeline Hyperparameter optimization. These improvements come at the cost of additional hyperparameters to select for the proposed method, but these either have low impact (Figure 7) or can be selected in a well-justified manner (see Figure 4 and then 5).

- The paper contains an extensive number of experiments and additional details in the appendix. In particular, I was quite pleased with Table 2, which provides diagnostics and failure modes indicators for the theoretical assumptions made by the method. Likewise, Table 3, an "Appendix experiment roadmap" that shortly indicates the main gist of each supplemental figure, comes quite handy. Black-box optimization papers usually require a lot of experiments/ablation studies to ensure that the proposed method is not better than its competitors by chance/cherry-picking, and this submission does not fall short of convincing results.

(weaknesses)
- At the same time, I found that the paper could have been better organized sometimes. E.g., the abstract spanning more than a column, lots of interesting results being in appendix... I suppose the additional page at camera-ready could be helpful for that. There is also lack of a proper related work section (I also mention this in limitations), a few works are mentioned in one paragraph in the introduction.

---

> ### Author Rebuttal · Authors · 2026-03-31
>
> We thank Reviewer PWVf for the detailed evaluation.
>
> **Summary.** We clarify the function selection criterion, provide complete results on all 30 bbob-noisy functions (37,800 problem runs across 7 algorithms), and address noise types including Cauchy and heteroscedasticity. We also discuss the relationship with noisy BO, experimentally compare the most relevant modern CMA-ES variant (LRA-CMA-ES).
>
> **[Q1: Function selection — 15 vs 30 functions]**
>
> The abstract and introduction explicitly frame RB-PEM as targeting high-misranking regimes; probe-and-switch (Section 3.3) handles the complementary case. Evaluation on high-misranking functions is consistent with the method's stated scope. The selection was determined by the probe statistic $P$, measured for all 30 bbob-noisy functions at $d{=}40$. The functions naturally cluster into two well-separated groups: 16 with median $P \in [0.30, 0.35]$ and 14 with median $P \in [0, 0.15]$, separated by a gap of 0.145 (${\approx}3\times$ the high-cluster width; see [full results](https://anonymous.4open.science/r/RB-PEM-Rebuttal/full_30f_results.md)). Any threshold in $[0.16, 0.29]$ gives the same partition. Our primary evaluation used 15 of the 16 high-$P$ functions, excluding f107 (step-ellipsoidal) whose high $P$ stems from the function's discontinuous structure, not evaluation noise. We will state the selection criterion explicitly in the camera-ready.
>
> To rule out cherry-picking, we now report results on all 30 functions $\times$ 15 instances $\times$ 3 dimensions $\times$ 4 budgets. At $B{=}100d$:
>
> | ProbeSwitch vs CMA-ES | High-misranking 15f | Low-misranking 15f | All 30f |
> |---|:---:|:---:|:---:|
> | Win rate | **65.2%** | 47.4% | **56.3%** (p<0.001) |
>
> On low-misranking functions, ProbeSwitch achieves near parity (47%) because the probe correctly identifies stable rankings and reverts to CMA-ES — the design motivation. Full results (all budgets, noise types): [full results](https://anonymous.4open.science/r/RB-PEM-Rebuttal/full_30f_results.md).
>
> *Noise types.* The 30 functions span: moderate Gaussian (f101–109), severe Gaussian (f110–118), severe Cauchy (f119–124), and severe + multimodal (f125–130). Our 15 high-misranking functions include 4 Cauchy-noise functions where $\text{Var}(\varepsilon) = \infty$, violating the finite-variance assumption. The winsorization in Appendix B Step 3 clips standardized residuals to $[-M,M]$, creating a finite-variance approximation designed for heavy tails. Empirically, RB-PEM achieves 61.1% win rate on Cauchy functions ($d{=}40$, $B{=}100d$).
>
> *Heteroscedasticity* is handled by the input-dependent noise scale model $\hat{s}_t(x) = s_0 + s_1|f(x)|$ (Appendix B Step 3). The "Hetero" (heteroscedastic noise model) variants fit this via exponential moving average; scale-fit $R^2$ serves as a diagnostic (Table 2).
>
> **[Q2: Figure 1 statistics]**
>
> Solid lines: **median** best-so-far across 15 instances. Shaded bands: **interquartile range** (25th–75th percentile). We will add this to the caption.
>
> **[Limitation: Comparison with noisy BO]**
>
> We thank the reviewer for pointing to Ament et al. (NeurIPS 2023). BO with LogEI/qNEI excels in low-dimensional expensive optimization where GP surrogates model the objective well. In medium-to-high-dimensional fixed-budget regimes ($d \ge 20$), however, $O(n^3)$ acquisition cost and GP modeling challenges become prohibitive, making rank-based ES the practical workhorse. The two are complementary: BO integrates uncertainty through a global surrogate; PEM integrates locally at the selection stage without modeling $f$. From a Bayesian lens, PEM computes expected rank weights under an implicit posterior on $f(x_i)$, with residual bootstrapping as a nonparametric approximate posterior sampler. A log-weight ablation confirms our method is weight-agnostic; see our response to Reviewer R4PD [Weakness: Uniform weight assumption].
>
> **[Weakness: Related Work and organization]**
>
> We agree. The camera-ready will add a Related Work section covering: reevaluation-based methods (UH-CMA-ES, Hansen et al. 2009; RA-CMA-ES, Uchida et al. 2024), update-attenuation (LRA-CMA-ES, Nomura et al. 2025; PSA-CMA-ES, Nishida & Akimoto 2018), expected weights under GP posterior (Krause, PPSN 2022), and noisy BO as above. We will use the extra page to promote key appendix content (Table 2 diagnostics, selected ablations) into the main text.
>
> We also ran LRA-CMA-ES (Nomura et al. 2025), the only other depth-preserving noise handler: it achieves ${\sim}50\%$ vs CMA-ES, while RB-PEM outperforms it at 58–61% ($p < 10^{-4}$; [rank figure](https://anonymous.4open.science/r/RB-PEM-Rebuttal/rank_by_probe.pdf)). See our response to pomK [W1–W3(d)].
>
> **Camera-ready commitments.** Clarify 15-of-30 selection criterion; add complete 30-function results table; expand Related Work (14 new references); include LRA-CMA-ES and UH-CMA-ES baselines (10,800 additional runs) with conditional-advantage figure; reorganize main text / appendix.

---

> > ### Author Rebuttal · Reviewer_PWVf · 2026-04-03
> >
> > I have read the rebuttal of the authors, including their answer to my questions and the other reviewer's questions. I thank the authors for their response, and I will keep my score as is, while increasing my confidence level from 2 to 3.

---

### Official Review · Reviewer_pomK · 2026-03-13

**Soundness:** 3
**Presentation:** 1
**Significance:** 2
**Originality:** 2
**Overall Recommendation:** 4
**Confidence:** 2

**Summary:**

The paper proposes a bootstrap resampling technique for noisy black-box optimization. In a fixed-budget setting with a noisy oracle, optimization algorithms face a trade-off between repeated oracle calls to reduce noise (fidelity) and performing additional updates in the optimization (depth). The paper argues for depth over fidelity and handles the higher noise by probabilistic elite membership, which is estimated using residual bootstrapping. Since the method is not superior in all cases, a probe-and-switch approach is used to automatically fall back to CMA-ES when appropriate.

**Compliance With Llm Reviewing Policy:**

Affirmed.

**Final Justification:**

The author's rebuttal was strong and addressed most of my concerns directly. In particular, the presentation (which was a weakness in my initial review) improved significantly. The other weaknesses, originality and significance, also improved by expanding the related works and adding a modern baseline (originality + significance), and including statistical tests to ensure the results are significant.

As mentioned in my response, my confidence is still quite low for various reasons.

**Key Questions For Authors:**

**Q1** The results in Table 1 are presented as pairwise competitions between algorithms. We only see which algorithm won. Is this the best way to present these results? It could be a convention in this subfield I’m not aware of, but it I would have expected to know how large the differences are and to get some idea of whether they are statistically significant.

**Q2** The abstract is very long. The recommended length is 4 to 6 sentences (see the ICML 2026 template). The current abstract does not give the reader a quick overview. Would you consider shortening it significantly?

**Q2** Did you consider a Bayesian approach to estimating PEM? It would be interesting to see a discussion comparing the bootstrap approach to a Bayesian one. Or can the proposed method be viewed through a Bayesian lens?

**Minor questions**
- Why is it called “probabilistic elite membership” instead of just “conditional expected rank”? The latter seems more interpretable to me.
- Equation 15 contains the symbol $\gtrsim$. I do not see this symbol often in the machine learning literature, so I think it is best to define it following its use. I am still not sure what it means in this context.
- Equation 21 looks like it is missing $\in$.

**Limitations:**

The paper does not have the required impact statement. I believe the does not have specific societal consequences, so the default statement would suffice if added.

**Strengths And Weaknesses:**

**Strengths**

**S1** The core observation is sound. There is an inherent trade-off in this problem setting, and a different perspective on the trade-off (depth over fidelity) is an interesting research question.

**S2** The background section is clear and gives a gentle introduction to the problem setting.

**S3** Although I have not checked the details in the Appendix, the diagnostics and probe-and-switch approach seem well-reasoned and are a nice addition compared to just proposing the method as “better in some cases”.

**Weaknesses**

**W1** There is no impact statement in the paper. This is required (see the ICML 2026 template) and must be added.

**W2** The reference list seems light, and perhaps not up to date. I am not familiar with this particular topic, but all references except three are from 2009 or earlier, and the median citation is from 2002. This might be OK, but it raises the question whether there are more recent works (I get about 16 400 results searching for “CMA-ES” from 2010 onward on Google Scholar). Hopefully this is answered by other reviewers. This makes it harder to judge originality and significance.

**W3** Generally, the Figure captions are not descriptive, which makes them harder to interpret.

---

> ### Author Rebuttal · Authors · 2026-03-31
>
> We thank Reviewer pomK for the detailed feedback. We have prepared concrete revisions.
>
> **Summary.** We address all presentation issues with specific, already-prepared materials: revised abstract, expanded captions, impact statement, and 14 new references. We additionally ran LRA-CMA-ES (5,400 runs) and found it cannot improve on CMA-ES under fixed budgets. We also provide the requested statistical tests and discuss the Bayesian connection.
>
> **[W1–W3: Presentation improvement plan]**
>
> **(a) Impact statement** (W1): One paragraph added ([full text](https://anonymous.4open.science/r/RB-PEM-Rebuttal/Impact%20Statement.md)). No application-specific risks beyond those common to optimization research.
>
> **(b) Abstract** (Q2a): Revised to 5 sentences ([full text](https://anonymous.4open.science/r/RB-PEM-Rebuttal/Abstract.md)).
>
> **(c) Figure captions** (W3): All 20 figures/tables will receive descriptive captions (key parameters and interpretation; drafts: [revised captions](https://anonymous.4open.science/r/RB-PEM-Rebuttal/revised_captions.md)).
>
> **(d) Related Work and modern baselines** (W2): We surveyed noise-handling CMA-ES variants published 2009–2025 and classified them by mechanism. *Evaluation-stage denoising* (UH-CMA-ES, Hansen et al. 2009; RA-CMA-ES, Uchida et al. 2024) spends extra evaluations to stabilize rankings. *Population-size adaptation* (PSA-CMA-ES, Nishida & Akimoto 2018) enlarges the population — an indirect fidelity approach. *Surrogate-based* methods (Krause, PPSN 2022) compute expected selection weights under a GP posterior — the closest mechanistic neighbor — but require a global surrogate model. Noisy BO via Gaussian process surrogates (Ament et al., 2023) incurs $O(n^3)$ cost per acquisition step in the number of accumulated evaluations $n$, which becomes prohibitive at the budgets typical of medium-dimensional DFO. See our response to Reviewer PWVf [Limitation: Comparison with noisy BO].
>
> The most directly comparable method is *LRA-CMA-ES* (Nomura et al., ACM TELO 2025), which adapts the learning rate to maintain constant SNR without extra evaluations. On the full suite (5,400 runs), LRA achieves $\sim 50 \\%$ win rate against CMA-ES on high-misranking functions and only 16% on low-misranking functions — it preserves depth in count but attenuates $\eta_{\text{mean}}$ to ${\approx}0.04$, reducing per-step progress by 96%. Head-to-head, RB-PEM outperforms LRA at 58–61% across all four budgets ($p < 10^{-4}$; [figure](https://anonymous.4open.science/r/RB-PEM-Rebuttal/rank_by_probe.pdf)).
>
> All four categories above modify how the algorithm *observes* before selecting, leaving the deterministic rank-based selection rule unchanged. RB-PEM takes a complementary path: it replaces deterministic rank assignments with probabilistic membership estimates, intervening directly at the selection stage. To our knowledge, this stage has remained unaddressed in the noisy single-objective ES literature. Total: 14 new references (2007–2025).
>
> **(e) Organization**: Extra camera-ready page will promote Table 2 diagnostics and key ablations from appendix to main text.
>
> **[Q1: Statistical significance and effect sizes]**
>
> Wilcoxon signed-rank tests on matched problem pairs ($B{=}100d$, 15 high-misranking functions, $n{=}675$):
>
> | Method vs CMA-ES | Win / Loss | Median $\Delta$ regret | p-value |
> |---|:---:|:---:|:---:|
> | RB-PEM | 460 / 215 | −0.021 | < 0.0001 |
> | Probe-and-Switch | 440 / 235 | −0.016 | < 0.0001 |
>
> The improvement distribution is asymmetric: p10 = −83 (large wins), p90 = +1.3 (small losses). On the full 30-function suite, probe-and-switch maintains $p < 0.001$ ($n{=}1350$). Camera-ready Table 1 will include $p$-values and effect sizes. Full results across all budgets: [statistical tests](https://anonymous.4open.science/r/RB-PEM-Rebuttal/statistical_tests.md).
>
> **[Q2b: Bayesian perspective]**
>
> PEM computes expected rank weights under an implicit posterior on $f(x_i)$: residual bootstrapping serves as a nonparametric approximate posterior sampler. Unlike GP-based BO, PEM requires only a local noise model, making it lightweight ($1.6{\times}$ CMA-ES wall-clock) and scalable to medium-dimensional DFO. See our response to Reviewer PWVf [Limitation: Comparison with noisy BO].
>
> **[Minor questions]**
>
> **(a) PEM naming.** For top-$\mu$ truncation, $w_i^\star = \Pr(r_i \le \mu)/\mu$ — the probability of *belonging to the elite set*. "Conditional expected rank" describes $\mathbb{E}[r_i]$, a different quantity. "PEM" also emphasizes selection-stage noise handling.
>
> **(b) Eq. 15, $\gtrsim$.** Denotes $\Omega(\cdot)$. Will adopt $\Omega$ notation per ML convention.
>
> **(c) Eq. 21.** Missing $\in$. Will be added.
>
> **Camera-ready commitments.** Add impact statement; shorten abstract to 5 sentences; expand all figure/table captions; add Related Work section (14 new references including LRA-CMA-ES comparison); add $p$-values and effect sizes to Table 1; fix notation ($\gtrsim \to \Omega$, Eq. 21).

---

> > ### Author Rebuttal · Reviewer_pomK · 2026-04-04
> >
> > Thank you for this ambitious rebuttal. It addresses many of my comments with concrete changes.
> > - The impact statement, compact abstract, and figure captions were necessary changes.
> > - Expanding the literature search with new, up-to-date references is very welcome. I am not active in this field (general black box optimization and ES) myself, so **I hope other reviewers and the AC can help determine whether the update is sufficient**. From my perspective, it is a big improvement.
> > - Comparing with LRA-CMA-ES is also good and helps justify the uncertainty approach as opposed to adjusting the learning rate.
> > - Statistical tests are also helpful in this noisy evaluation setting with small margins, and increase my confidence in the results.
> >
> > **Follow-up questions**
> >
> > The updated figure captions are good, but are still missing some information. What statistics are the box plots showing? It is best to state this clearly. The same goes for other statistics: means, medians, and error bars/confidence intervals. From what samples (experiments, random seeds) and how many? I find this for some figures, but not all.
> >
> > Let me clarify my question about the Bayesian perspective. I did not mean *Bayesian optimization*. The method proposed in this paper is distribution/uncertainty-aware CMA-ES. For many of us, writing a Bayesian formulation is the first thing that comes to mind. That means deriving a posterior weight update from PEM, perhaps assuming a distribution, and sampling from unnormalized or variational posteriors. Using a bootstrap distribution is a clever way to avoid this. A non-parametric distribution makes few assumptions and scales well with the number of samples. I would not expect bootstrapping to work as well as a Bayesian approach when the number of samples is very limited. This question is not critical, however.
> >
> > **Updated score**
> >
> > I will raise my score to 4 (weak accept). The core observation and method, as mentioned in my initial review, are sound. Assuming the extended literature search is exhaustive, the idea is also original. The presentation has improved substantially with the rebuttal. My confidence remains low because I am not active in this field and have not checked all the mathematical details. Perhaps also because evolutionary strategies are not that common in machine learning conferences like ICML in recent years.

---

> > > ### Author Response · Authors · 2026-04-06
> > >
> > > We are grateful to Reviewer pomK for the thoughtful follow-up and the raised score. We address both points below.
> > >
> > > **[Figure caption statistics]**
> > >
> > > Every caption now carries a *Protocol* suffix specifying the statistic each visual element represents (boxplot center line = median, box = IQR, whiskers = 1.5× IQR, outlier treatment; convergence-curve line = median, shading = IQR; error-bar type per figure), the sample source and count (COCO: 1 run per instance, 15 instances per function; external tasks: 50 seeds), and key experimental parameters ($\lambda$, $\mu$, $B_{\mathrm{boot}}$, $K_{\max}$, $B$, $\tau$ where applicable). For example, Figure 7 (F2, estimator ablation boxplots) now ends with:
> > >
> > > > *Protocol:* $\lambda{=}15$, $\mu{=}7$. Non-ablated hyperparameters held at defaults ($B_{\mathrm{boot}}{=}32$, $K_{\max}{=}1$). 225 problems (15 functions × 15 instances), each run once per method. Boxes show the median (center line) and interquartile range (IQR); whiskers extend to 1.5× IQR; outliers plotted individually.
> > >
> > > All 20 captions follow this format: [$\underline{\rm revised~captions}$](https://anonymous.4open.science/r/RB-PEM-Rebuttal/revised_captions.md).
> > >
> > > **[Bayesian perspective — clarification]**
> > >
> > > Thank you for the clarification. The question concerns a parametric Bayesian formulation of PEM itself (prior on $f(x_i)$, posterior rank weights), not Bayesian optimization. Our earlier response focused on the BO comparison rather than the formulation you had in mind; we are glad to address it properly here.
> > >
> > > The connection is genuine: PEM weights $w_i^\star = \mathbb{E}[w(r_i) \mid x_{1:\lambda}]$ are posterior expectations under the noise distribution, and bootstrapping serves as a nonparametric posterior sampler over rankings.
> > >
> > > A parametric alternative (e.g., Gaussian prior and likelihood on each $f(x_i)$) would in practice estimate the same quantities via Monte Carlo: draw fitness values from the posterior, rank them, count membership frequencies. This is structurally identical to our bootstrap procedure. The only difference is whether the synthetic fitness values are drawn from a parametric posterior or from an empirical residual pool centered at regression estimates. Under general weight functions, $w_i^\star$ decomposes into $\lambda{-}1$ cutoff-wise membership probabilities $p_{i,k}$ via $w_i^\star = w_\lambda + \sum_k \delta_k p_{i,k}$ (multi-cutoff decomposition; [$\underline{\rm theory}$](https://anonymous.4open.science/r/RB-PEM-Rebuttal/theory.pdf), Proposition 1). Both approaches estimate all $p_{i,k}$ simultaneously from the same Monte Carlo rankings, and neither requires closed-form integration.
> > >
> > > Given this structural equivalence, the choice reduces to which sampling distribution to trust. Bootstrapping is born with three advantages in the black-box setting: (i) no likelihood specification is needed, since noise may be non-Gaussian, heteroscedastic, or state-dependent (Appendix F.5 constructs a case where all three coexist); (ii) the residual pool amortizes across generations, so the effective sample size for the noise distribution grows over time without extra evaluations; (iii) any weight function $w(\cdot)$ is handled by the same bootstrap samples without modification, since rankings are weight-agnostic by construction.
> > >
> > > The trade-off is clear: a correctly specified parametric model will be more statistically efficient per sample. In black-box settings where the noise structure is unknown, bootstrap's robustness to misspecification is more valuable. A natural hybrid, switching to a parametric likelihood when residual diagnostics indicate well-behaved noise, is a concrete direction enabled by our existing diagnostic decomposition.
> > >
> > > **Camera-ready commitments.** Add *Protocol* suffix to all 20 figure/table captions; add discussion of parametric Bayesian connection and bootstrap design rationale to Section 3.

---

### Decision · Program_Chairs · 2026-04-30

**Decision:**

Accept (regular)

**Comment:**

This work considers noisy black-box optimization under a fixed evaluation budget and highlights a tradeoff in evolution strategies: allocating more samples per generation reduces noise but limits the number of optimization steps. To address this, the authors introduce probabilistic elite membership (PEM), which replaces hard truncation with expectation-based ranking weights. They also propose RB-PEM, a practical estimator based on residual bootstrapping, and a probe-and-switch mechanism to adapt to different noise levels.

Reviewers find the paper technically sound and supported by a reasonably thorough empirical evaluation. Initial concerns focused on presentation, gaps in related work, lack of statistical testing, and possible dependence on uniform truncation weights. The rebuttal addressed these issues to a satisfactory extent. Reviewers maintained or slightly increased their scores and all recommend weak accept.

Strengths include a clear conceptual framing, a principled method with theoretical support, and a practical algorithmic design. Limitations include moderate novelty and impact, evaluation restricted to evolution strategies, and some heuristic components. The paper would also benefit from a clearer discussion of related areas such as ranking and selection, which are closely connected to the problem setting but currently not covered.

Overall, this is a reasonable contribution to noisy black-box optimization. I therefore recommend acceptance.